# MreB polymers and curvature localization are enhanced by RodZ and predict *E. coli*'s cylindrical uniformity

Benjamin P. Bratton [1,2], Joshua W. Shaevitz[2], Zemer Gitai [1] & Randy M. Morgenstein [1,3]

The actin-like protein MreB has been proposed to coordinate the synthesis of the cell wall to determine cell shape in bacteria. MreB is preferentially localized to areas of the cell with specific curved geometries, avoiding the cell poles. It remains unclear whether MreB's curvature preference is regulated by additional factors, and which specific features of MreB promote specific features of rod shape growth. Here, we show that the transmembrane protein RodZ modulates MreB curvature preference and polymer number in *E. coli*, properties which are regulated independently. An unbiased machine learning analysis shows that MreB polymer number, the total length of MreB polymers, and MreB curvature preference are key correlates of cylindrical uniformity, the variability in radius within a single cell. Changes in the values of these parameters are highly predictive of the resulting changes in cell shape ($r^2 = 0.93$). Our data thus suggest RodZ promotes the assembly of geometrically-localized MreB polymers that lead to the growth of uniform cylinders.

[1] Department of Molecular Biology, Princeton University, Princeton, NJ 08544, USA. [2] Department of Physics and Lewis-Sigler Institute for Integrative Genomics, Princeton University, Princeton, NJ 08544, USA. [3] Department of Microbiology and Molecular Genetics, Oklahoma State University, Stillwater, OK 74078, USA. Correspondence and requests for materials should be addressed to Z.G. (email: zgitai@princeton.edu) or to R.M.M. (email: randy.morgenstein@okstate.edu)

Understanding how cells encode the ability to robustly determine their own shapes remains one of the central mysteries of cell biology. In bacteria, the peptidoglycan cell wall (PG) forms a rigid structure whose shape dictates the shape of the cell. When purified, the extracted PG maintains the cell's original shape and loss of PG causes cells to lose their shapes, e.g., rod-shaped bacteria becoming round[1–3]. These cells can then reestablish cell shape de novo once cell wall synthesis is restored[2]. In this work, we focus on the gram-negative rod-shaped bacterium *Escherichia coli*. Despite their relative simplicity, there are multiple parameters that describe a population of rod-like cells including: a straight cylindrical axis with high uniformity of the diameter (which we term cylindrical uniformity), centerline straightness (bent vs. straight rod), the distribution of cell widths and lengths within the population, and the geometry of the poles. Previously we studied *E. coli* straightening and population-average width, but the MreB properties that correlate with single-cell cylindrical uniformity remain unclear[2,4,5]. Another unresolved question concerns which properties of MreB are intrinsic and which are regulated by accessory factors.

In *E. coli*, cell wall insertion is localized to the main body of the cell, with no growth at the poles where the cell wall remains inert[6]. This cell wall insertion pattern mirrors the geometric localization pattern of MreB, which is depleted from the poles and is enriched at areas of low or negative mean curvature[4]. Oriented MreB polymer motion can cause MreB to spend less time at the poles[7]. In mechanically strained cells geometric localization may be insufficient to explain cell straightening[8]. Because the scale of cellular curvature is much larger than that of a single protein, it is difficult for monomeric proteins to detect the difference between polar and mid-cell geometry[9,10]. To overcome this problem, MreB has been proposed to form cellular-scale polymers, whose assembly has been observed both in vivo and in vitro[11–14]. In vitro, MreB filaments can bend a membrane vesicle and molecular dynamics simulations suggest that MreB polymers have an intrinsic bend[12,15].

MreB polymers have been proposed to serve multiple roles in cell shape determination[16]. First, the twisting cylindrical growth seen in *E. coli* is disrupted by treatment with A22, an inhibitor of MreB assembly[17,18]. Second, the orientation of MreB polymers relative to the cell axis is correlated with the average cell width of the population[5]. However, previous studies have not examined the MreB properties coupled to the ability to form an elongated cylindrical rod-like cell.

Several toxins have been proposed to target MreB under conditions of stress[19–21], but it remains unclear whether MreB assembly or curvature localization are normally regulated in *E. coli*. RodZ is a transmembrane protein (Fig. 1a) that is co-conserved with MreB[22] and is one of the few proteins that definitively binds MreB, as shown in vitro by co-crystallization and in vivo by bimolecular fluorescence complementation (BiFC)[1,23]. We previously showed that RodZ functions downstream of MreB as an adapter that enhances MreB circumferential rotation by promoting the association of cytoplasmic MreB to the periplasmic activity of cell wall synthesis[1].

Here we show that RodZ is also a key regulator of MreB assembly and curvature preference. These functions require both the cytoplasmic and periplasmic domains of RodZ, indicating that RodZ functions as a key hub to integrate information across the inner membrane and organize cell shape. Using three-dimensional (3D) imaging and a combination of *mreB* and *rodZ* mutants, we go on to use an unbiased machine learning approach to explore which of the many properties of MreB are predictive of cylindrical uniformity. We find that a combination of the changes in MreB polymer number, total polymer length, and curvature preference accurately predict changes in cylindrical uniformity.

## Results

**RodZ is required for MreB curvature localization.** Since RodZ has emerged as a central coordinator of MreB function, we examined the role of RodZ in controlling the biophysical properties of MreB that are thought to be important for shape determination, like curvature preference[24]. To quantify the effect of RodZ on MreB curvature preference we measured the 3D cell shape and curvature enrichment of MreB in a strain expressing MreB-GFP^sw (internal msGFP sandwich fusion) as the sole copy of MreB (Fig. 1b). We previously showed that this fusion fully complements the shape of WT *E. coli* under a wide range of conditions[5] and all mutants described below were generated in this strain background. Generating 3D cell-shape reconstructions with roughly 50 nm precision from the raw fluorescence images allowed us to calculate the Gaussian curvature, which is the product of the two principal curvatures, at every location on the 3D surface of the cell[25]. These two principal curvatures can only be measured in 3D. Previously we focused on MreB's curvature preference as a function of mean curvature[4], the average of the two principal curvatures. Mean curvature is sensitive to global properties such as cell size, whereas Gaussian curvature enables us to focus on the local curvature geometry, which is particularly important in irregularly-shaped cells such as Δ*rodZ* mutants.

Because the absolute concentration of MreB can vary between cells, we set the average MreB concentration for each individual cell to one and measured that cell's MreB curvature-dependent concentration relative to that average value, normalized by the amount of that curvature available. We then averaged these single cell measurements across multiple cells to obtain an enrichment/depletion profile. Enrichment/depletion values of one indicate that the average MreB concentration at that curvature is the same as the average concentration of MreB across the cell surface while values above one indicate curvatures where MreB is enriched and values below one indicate curvatures where MreB is depleted. In WT-cells, MreB is enriched at negative and low positive Gaussian curvatures (including zero) and depleted from high positive Gaussian curvature (Fig. 1c–e).

These curvature enrichment profiles are consistent with previous reports that negative Gaussian curvature values show enriched MreB localization (enrichment/depletion values >1), and further show that MreB is also enriched at zero and small positive Gaussian curvature. We note that in some conditions there is a peak in MreB enrichment near zero Gaussian curvature. This peak is not seen in all conditions and, regardless of whether there is a peak, MreB remains enriched at negative Gaussian curvature because the enrichment profile does not fall below one at negative curvatures. Cell poles have relatively high positive Gaussian curvature since each of the principal curvatures at the pole have the same sign, while cylinders have a Gaussian curvature of zero owing to the lack of curvature along the cell axis. Thus, MreB's curvature preference nicely parallels the pattern of *E. coli* growth during elongation, localizing to cylindrical regions and avoiding the poles.

Interestingly, we found that deletion of *rodZ* strongly reduced the curvature preference of MreB (Fig. 1c–e). In Δ*rodZ* cells, MreB is no longer enriched near zero Gaussian curvature or excluded from the poles. The shape of Δ*rodZ* cells can be complemented by expressing full-length RodZ from a plasmid (RodZ$_{1–337}$) (Fig. 1b, Supplementary Fig. 1). RodZ$_{1–337}$ also restores both the depletion of MreB from regions of positive Gaussian curvature and the enrichment of MreB in regions of negative Gaussian curvature. These cells lacked the peak in MreB

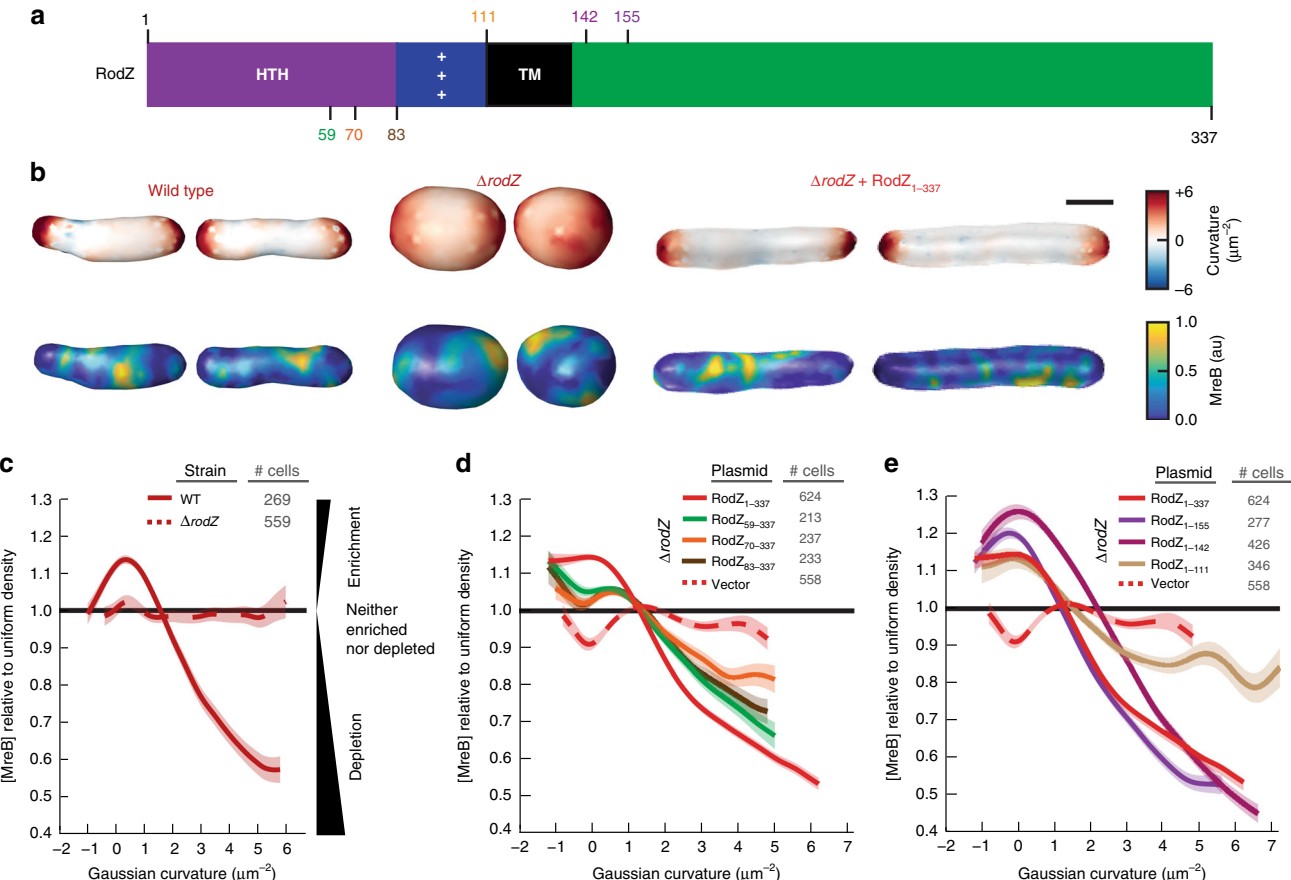

**Fig. 1** The cytoplasmic domain of RodZ is necessary for MreB curvature localization. **a** Schematic of RodZ's domain structure with location of truncations noted. **b** 3D images of WT, $\Delta rodZ$, and $rodZ$ complemented cells with Guassian curvature and MreB fluorescence intensity represented. **c** MreB enrichment plot of WT and $\Delta rodZ$. Values >1 show enrichment and <1 show depletion relative to uniform coverage of the cell. **d** MreB enrichment plot of RodZ cytoplasmic truncations. **e** MreB enrichment plot of RodZ periplasmic truncations. Shaded area indicates ± 1 standard error of the mean, and dotted lines indicate strains deleted of $rodZ$. **c–e** The curve for each strain is a cubic smoothing spline and is truncated using a probability threshold for extreme curvatures of $p > 5 \times 10^{-3}$ (Supplementary Figs. 4 and 5). Because the shape of each strain is different, the ranges of curvatures plotted for each strain are different. These experiments were performed on three separate days and the data were pooled

enrichment near zero Gaussian curvature noticeable in WT cells (Fig. 1c) but clearly retained depletion of MreB from positive Gaussian curvature and enrichment at low-to-negative Gaussian curvature.

Because $\Delta rodZ$ disrupts both curvature localization and rod shape, we sought to determine if RodZ's affect of MreB curvature is secondary to its affect on cell shape. We thus grew WT cells in sub-lethal concentrations of the PBP2-inhibiting drug, mecillinam. Mecillinam treatment disrupted cylindrical uniformity to a similar extent as loss of $rodZ$ (Supplementary Fig. 1). But in contrast to $\Delta rodZ$ cells, mecillinam-treated cells maintained MreB's geometric localization with preference for MreB to localize to Gaussian curvatures near and below zero and avoid high positive Gaussian curvatures (Supplementary Fig. 2a–c). As another way to uncouple cell shape and MreB curvature enrichment we analyzed MreB's curvature enrichment after washout of A22. Growth in A22 disrupted cylindrical uniformity to a similar extent to mecillinam treatment and $\Delta rodZ$ (Supplementary Fig. 1). After washout, MreB filaments reassemble rapidly (minute-timescale), while rod shape recovers slowly (hour-timescale). Imaging 5–10 min after A22 washout thus allowed us to examine assembled MreB in shape-disrupted cells and confirmed that despite their irregular shapes these cells maintained geometrically-localized MreB (Supplementary Fig. 3). Because mecillinam-treated, A22-treated, and $\Delta rodZ$ cells all lack

rod-like shape but only $\Delta rodZ$ cells lack geometrically localized MreB, the lack of MreB enrichment in $\Delta rodZ$ must not be a failure in our 3D analysis nor a result of changes in cellular geometry. These data show that RodZ specifically promotes MreB's curvature localization in a manner that is not merely secondary to its role in cell shape determination.

**RodZ's cytoplasmic domain tunes MreB curvature localization.** RodZ is a transmembrane protein with a large periplasmic domain and a smaller cytoplasmic domain (Fig. 1a). We hypothesized that these two domains of RodZ could play distinct roles, with the periplasmic domain binding the PG synthesis machinery to promote MreB rotation, and the cytoplasmic domain binding MreB to promote its curvature preference. In order to determine how RodZ regulates MreB curvature localization, we thus examined MreB curvature enrichment in RodZ truncations from both its periplasmic and cytoplasmic termini.

Consistent with our hypothesis, we find that the periplasmic domain plays little role in modulating MreB's curvature preference (Fig. 1e) even though it is necessary for cell shape[1,13,26]. For example, the curvature localization of RodZ$_{1-155}$ is largely indistinguishable from that of RodZ$_{1-337}$, and RodZ$_{1-142}$ retains geometrically-localized MreB. Even after deleting the entire periplasmic domain along with the

transmembrane domain (RodZ$_{1-111}$), there is still a noticeable enrichment around zero Gaussian curvature and a steady decline in enrichment as the Gaussian curvature becomes more positive. This indicates that even when RodZ is not in the membrane, its cytoplasmic domain can influence MreB's curvature preference. Unlike the membrane-bound periplasmic truncations (RodZ$_{1-155}$ and RodZ$_{1-142}$), a small truncation in the RodZ cytoplasmic domain (RodZ$_{59-337}$) shows a clear change in MreB curvature preference with both a decreased enrichment near zero and less of a depletion from regions with positive Gaussian curvature (Fig. 1d) and a concurrent loss of cell shape (Supplementary Fig. 4c–e). A further truncation (RodZ$_{70-337}$) and deletion of the entire helix-turn-helix motif (RodZ$_{83-337}$) show a further dampening of the curvature enrichment profile. While they fail to complement MreB curvature localization, cytoplasmic truncations do generate different cell shapes, suggesting that these truncations are being stably expressed. For all of the Gaussian curvature preference measurements reported in this study, we take into account the distributions of curvatures observed such that changes in curvature preference are not due to changes in the available curvatures in the cell (Supplementary Figs 2–5).

### Geometrically-localized MreB is not sufficient for rod shape.

Deleting *rodZ* results in a loss of cylindrical uniformity that can be suppressed by a point mutant in *mreB* (MreB$_{S14A}$). Because RodZ is needed for MreB's proper curvature localization, we determined whether MreB$_{S14A}$ can also suppress the loss of MreB's curvature enrichment in the absence of *rodZ*. We found that the curvature enrichment profile of MreB$_{S14A}$ in Δ*rodZ* shared many features with WT (Fig. 2a, b): both enrichment profiles are enriched at negative, zero, and weakly positive curvatures with a sharp decline toward depletion at strongly positive Gaussian curvature.

To test whether the correlation between shape and MreB localization observed for MreB$_{S14A}$ Δ*rodZ* is generalizable, we examined additional MreB point mutations that were originally identified as resistant to the MreB-targeting drug, A22[5]. We confirmed that the steady state levels of MreB are not dramatically affected by these point mutations in the presence or absence of *rodZ* (Supplementary Fig. 6). MreB$_{E143A}$ had little to no effect on MreB localization in the presence of RodZ while MreB$_{E143A}$ Δ*rodZ* restored geometric localization to MreB (Fig. 2a, b). Despite their qualitatively similar MreB curvature enrichment profiles, MreB$_{S14A}$ Δ*rodZ* formed rods that closely resembled WT cells while MreB$_{E143A}$ Δ*rodZ* cells were more irregular (Fig. 2b, Supplementary Data 1). Analysis of another point mutant, MreB$_{Y183N}$, reinforced the conclusion that proper curvature localization is insufficient for proper rod shape (Supplementary Fig. 7). MreB$_{Y183N}$ failed to form rods in the presence of RodZ, despite displaying geometrically-localized MreB (Fig. 2c). Thus, all the rod-shaped cells we analyzed have geometrically-localized MreB, but not all cells with geometrically-localized MreB form rods.

### RodZ modulates the number of MreB polymers.

Given that RodZ promotes both MreB rotation and curvature preference, we sought to determine if there are additional properties of MreB under RodZ control. Specifically, our 3D analysis enabled us to determine MreB polymer length, number, angle with respect to the long cell axis, and fraction of membrane-associated protein. Comparing cells with and without RodZ, we observed that there were not substantial changes in polymer length (Fig. 3) or average polymer angle (Supplementary Fig. 6a). There was a statistically-significant change in the fraction of MreB associated with the cell periphery (Supplementary Fig. 6d), but this change was small (~5%) and also observed in mecillinam-rounded cells, such that it does not appear to be a major component of RodZ's influence on

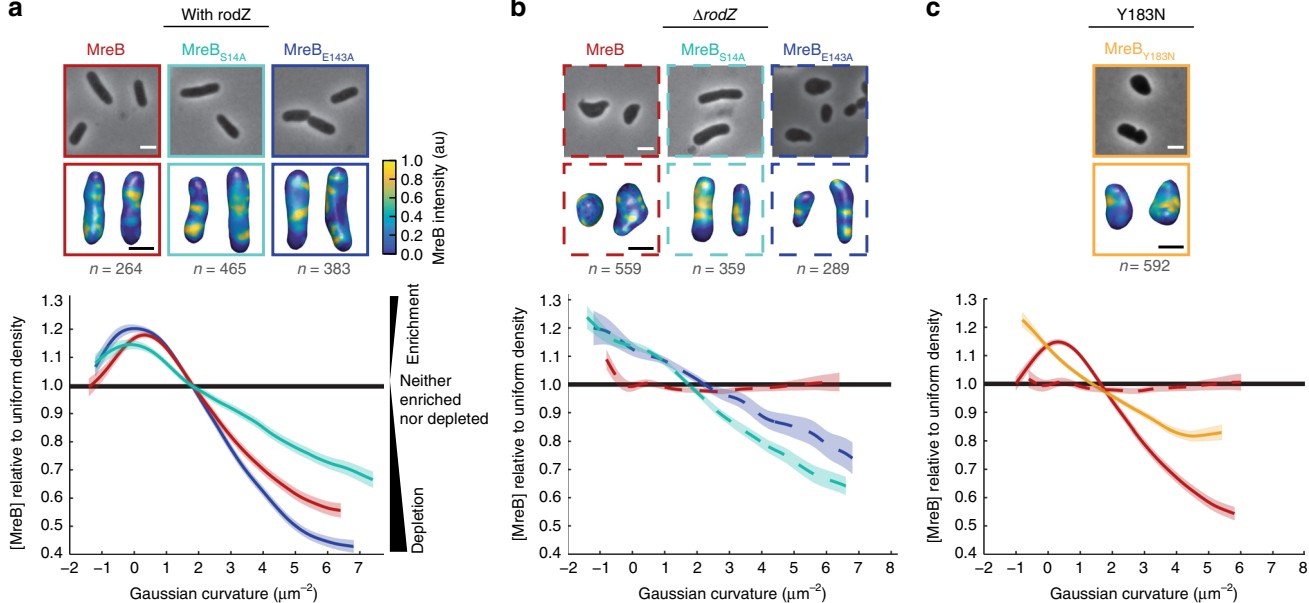

**Fig. 2** MreB curvature localization is necessary but not sufficient for rod shape determination. **a** MreB enrichment curves of MreB point mutants with RodZ present. **b** MreB enrichment curves of MreB point mutants in a *rodZ* deletion. **c** MreB enrichment curve of MreB$_{Y183N}$. Top images are 2D cells and bottom images are 3D cells with MreB shown according to the color intensity scale in **a**. The number of independent cells that contributed to the enrichment plots is indicated in gray. Shaded areas of the curves indicate ± 1 standard error of the mean and dotted lines indicate strains deleted of *rodZ*. The curve for each strain is a cubic smoothing spline and is truncated using a probability threshold for extreme curvatures of $p > 5 \times 10^{-3}$ (Supplementary Fig. 5). Because the shape of each strain is different, the ranges of curvatures plotted for each strain are different. These experiments were performed on three separate days and the data were pooled. White scale bar for all phase images is 2 μm and the black scale bar for all 3D reconstructions is 1 μm

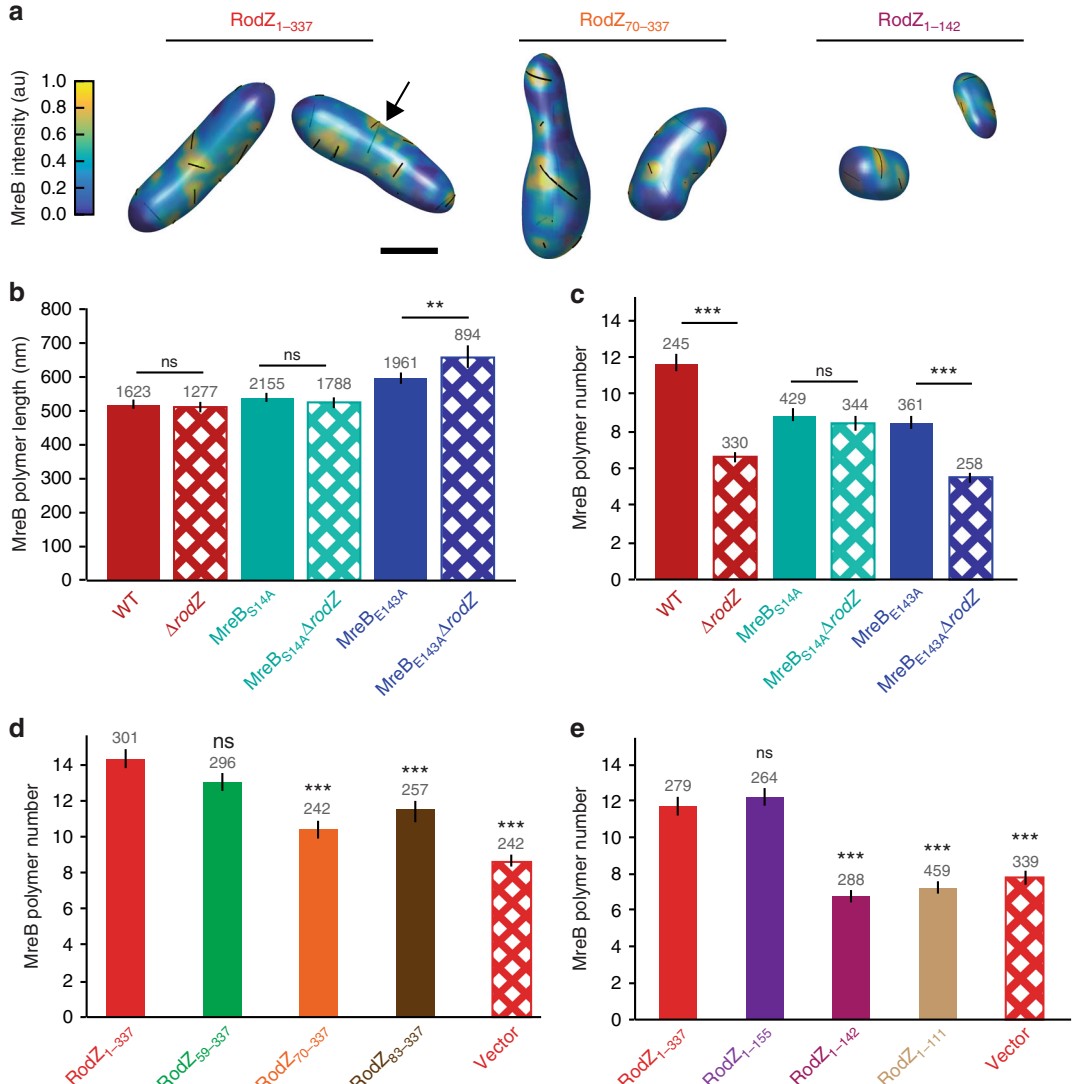

**Fig. 3** RodZ acts as an MreB assembly factor. **a** Semi-transparent 3D renderings of cells with full-length or truncated RodZ as indicated. MreB polymers are indicated with black lines and may be present on the back of the cell where they appear less vividly black (see example at arrow). **b** The average MreB polymer length (>200 nm) per cell in different MreB point mutants in the presence and absence of *rodZ*. **c** The average MreB polymer number per cell in different MreB point mutants in the presence and absence of *rodZ*. **b**, **c** *p*-value from Student's *t*-test comparisons are made between strains with similar MreB point mutations. For additional cross-comparisons see Supplementary Table 3. **d** The average MreB polymer number per cell in RodZ cytoplasmic truncation mutants. **e** The average MreB polymer number per cell in RodZ periplasmic truncation mutants. **b–e** These experiments were performed on three separate days and the data were pooled. *p*-value from Student's *t*-test comparisons are made between indicated strain and a strain with full length RodZ (solid red bar). For other comparisons see Supplementary Table 3. The number above the bars is the number of polymers (**b**) and the number of cells (**c–e**) analyzed. Error bars represent 95% confidence intervals. ns *p* > 0.05, **\*\***$p \leq 0.01$, **\*\*\***$p \leq 0.001$

MreB. We note that our forward convolution method[5] can accurately identify the presence of polymers regardless of their length and can determine the length of polymers greater than 200 nm. Thus, we can calculate the average MreB polymer length for the ~60% of polymers that are longer than 200 nm. When these MreB polymers are measured in strains with or without RodZ, there is not a significant difference in the average MreB polymer length (WT = $515 \pm 15$ nm, $\Delta rodZ = 509 \pm 18$ nm) (Fig. 3b), nor in the fraction of detected structures >200 nm (57%, 59%).

While RodZ does not influence all aspects of MreB polymers, we observed a dramatic decrease in total MreB polymer number per cell (including those <200 nm) in the absence of RodZ (WT = $11.6 \pm 0.5$, $\Delta rodZ = 6.6 \pm 0.3$) (Fig. 3c). The change in polymer number in $\Delta rodZ$ cells cannot be attributed to changes in cell shape as mecillinam treatment led to round cells without a

concurrent decrease in polymer number (polymer numbers actually increased in these cells, (Supplementary Fig. 2d). Furthermore, we binned both WT and $\Delta rodZ$ cells by volume and found that in cells of similar volume $\Delta rodZ$ had fewer MreB polymers and lacked geometrically-localized MreB (Supplementary Fig. 8). Thus, while cell volume does affect polymer number, RodZ increases the number of polymers and promotes curvature localization independently of its effect on cell size and shape.

To further dissect the role of RodZ in assembling MreB polymers, we examined the average polymer number in cells with RodZ truncation mutants. Because MreB curvature localization is more dependent on the cytoplasmic domain of RodZ than its periplasmic domain and MreB binds RodZ in the cytoplasm, we hypothesized that this domain would also control MreB polymer number. As expected, deletion of the cytoplasmic domain of

RodZ resulted in a decrease of the number of MreB polymers per cell (Fig. 3d). Surprisingly, we also saw a dramatic reduction in the number of MreB polymers per cell when we truncated the periplasmic domain of RodZ (Fig. 3e). Since the periplasmic domain of RodZ is needed to interact with the cell wall synthesis machinery, these data suggest RodZ could integrate signals from the process of cell growth to feed back on MreB and control polymer number.

We also compared MreB polymer properties in MreB point mutants with or without *rodZ*. We found that specific point mutations altered specific properties of MreB. For example, MreB$_{E143A}$ polymers are longer than WT but have the same MreB polymer angle, while MreB$_{S14A}$ polymers are the same length as WT but the polymer angle is different (Fig. 3a and Supplementary Fig. 6). Interestingly, when comparing MreB$_{S14A}$ in the presence or absence of RodZ, MreB$_{S14A}$ suppressed the RodZ-dependent properties of MreB (curvature localization, polymer number, and membrane-association) (Fig. 3a and Supplementary Fig. 6). MreB$_{S14A}$ was also the strongest suppressor of $\Delta rodZ$ cell shape, suggesting that MreB$_{S14A}$ functionally restores a majority of the effect of the WT MreB-RodZ interaction. In contrast, MreB$_{E143A}$, a partial suppressor of $\Delta rodZ$ cell shape, suppresses the effects of RodZ on MreB curvature localization but does not suppress the effects of RodZ on MreB polymer number or membrane fraction. MreB$_{E143A}$ also has longer polymers than MreB$_{WT}$ and the length of these polymers increases in the absence of RodZ. Together our results suggest that the different properties of MreB can be modulated independently.

**Specific MreB parameters predict growth as uniform rods**. Because MreB curvature preference did not always correlate with cylindrical uniformity and MreB parameters can be independently controlled, we sought to determine which properties of MreB best predict cylindrical uniformity. Specifically, we took a

machine learning approach to identify which features of MreB are most predictive of cell shape perturbations in an unbiased data-driven manner that is independent of any assumptions about the mechanism by which those features might function.

As inputs for our mechanism-independent machine learning procedure we quantified cell shape and compared MreB properties (Supplementary Table 2) across a large set of *mreB* and *rodZ* mutants (Supplementary Table 1, Supplementary Data 1). To quantify cylindrical uniformity we relied on our previous analysis[1] showing that the variation of cell diameter within a single cell (intracellular diameter deviation, IDD) is a quantitative measure of cylindrical uniformity (Supplementary Fig. 9). We confirmed that the IDD measured from 3D reconstructions also shows a clear separation between cells that are qualitatively classified as uniform rods, irregular rods, and round cells (note that small IDD corresponds to cylindrically uniform objects, Supplementary Fig. 9). We then built a collection of shape comparisons by computing the difference in IDD between two strains ($\Delta \text{IDD} = \text{IDD}_{strain1} - \text{IDD}_{strain2}$) (Table 1). Using this nomenclature, a positive $\Delta \text{IDD}$ describes a comparison where cells of strain 1 are more irregular in their shape than cells of strain 2. For example, $\Delta rodZ$ cells have a $\Delta \text{IDD}$ of $+0.1$ when compared to WT cells ($\Delta \text{IDD} = \text{IDD}_{\Delta rodZ} - \text{IDD}_{WT}$). We note that in all cases we computed a $\Delta \text{IDD}$ value that compares two strains with one change, either comparing the same genetic background with or without *rodZ* to assess the impact of RodZ, or comparing WT to different alleles of *mreB* to assess the impact of specific changes to MreB. In addition to $\Delta \text{IDD}$, we computed the change in MreB parameters for these same comparisons, choosing as our input a wide variety of scalar quantities (average polymer length, number of polymers, polymer angle, fraction of MreB on the membrane, etc.) and versions of these normalized by the surface area or volume. For the non-scalar metric (curvature localization), we distilled the curvature enrichment profiles into multiple scalars, including the average of the relative MreB

**Table 1 LASSO analysis of MreB's role in modulating cylindrical uniformity**

| Symbol[a] | Strain 1 | Strain 2 | Predicted shape change[b] | Observed shape change[c] |
|---|---|---|---|---|
| Training | | | | |
| □ | $\Delta rodZ$ | WT | 0.109 | 0.101 |
| □ | $\Delta rodZ + rodZ$ | WT | −0.022 | 0.000 |
| ▽ | $\Delta rodZ + rodZ_{59\text{-}337}$ | $\Delta rodZ + rodZ$ | 0.036 | 0.051 |
| ▽ | $\Delta rodZ + rodZ_{70\text{-}337}$ | $\Delta rodZ + rodZ$ | 0.082 | 0.076 |
| ▽ | $\Delta rodZ + rodZ_{83\text{-}337}$ | $\Delta rodZ + rodZ$ | 0.059 | 0.073 |
| △ | $\Delta rodZ + rodZ_{1\text{-}155}$ | $\Delta rodZ + rodZ$ | 0.012 | 0.005 |
| △ | $\Delta rodZ + rodZ_{1\text{-}142}$ | $\Delta rodZ + rodZ$ | 0.040 | 0.045 |
| △ | $\Delta rodZ + rodZ_{1\text{-}111}$ | $\Delta rodZ + rodZ$ | 0.077 | 0.070 |
| □ | $\Delta rodZ + \text{pTrc99A}$ | WT | 0.071 | 0.098 |
| ▽ | $\Delta rodZ + rodZ_{59\text{-}337}$ | $\Delta rodZ + \text{pTrc99A}$ | −0.062 | −0.047 |
| ▽ | $\Delta rodZ + rodZ_{70\text{-}337}$ | $\Delta rodZ + \text{pTrc99A}$ | −0.016 | −0.022 |
| ▽ | $\Delta rodZ + rodZ_{83\text{-}337}$ | $\Delta rodZ + \text{pTrc99A}$ | −0.039 | −0.025 |
| △ | $\Delta rodZ + rodZ_{1\text{-}155}$ | $\Delta rodZ + \text{pTrc99A}$ | −0.086 | −0.093 |
| △ | $\Delta rodZ + rodZ_{1\text{-}142}$ | $\Delta rodZ + \text{pTrc99A}$ | −0.057 | −0.053 |
| △ | $\Delta rodZ + rodZ_{1\text{-}111}$ | $\Delta rodZ + \text{pTrc99A}$ | −0.020 | −0.028 |
| □ | MreB$_{S14A}$ | WT | 0.020 | 0.013 |
| □ | MreB$_{E143A}$ | WT | 0.017 | 0.007 |
| □ | MreB$_{Y183N}$ | WT | 0.075 | 0.120 |
| □ | MreB$_{S14A}$ $\Delta rodZ$ | MreB$_{S14A}$ | 0.028 | 0.012 |
| □ | MreB$_{E143A}$ $\Delta rodZ$ | MreB$_{E143A}$ | 0.072 | 0.053 |
| Testing | | | | |
| • | MreB$_{E143A}$ $\Delta rodZ + \text{pTrc99A}$ | MreB$_{E143A}$ $\Delta rodZ + \text{MreB}_{E143A}$ | 0.027 | 0.045 |

[a]Symbols used in Fig. 4 to show the different types of genetic perturbations included in the regression
[b]Predicted change in cell shape ($\Delta \text{IDD} = \text{IDD}_{strain1} - \text{IDD}_{strain2}$) is predicted from the regression model
[c]Observed change in cell shape is from 3D measurements of cells

concentration for Gaussian curvatures below and above 2 μm$^{-2}$. For a complete list of MreB parameters quantified, see Supplementary Table 2. We note that this approach makes no assumptions about which MreB polymers are functional and merely determines which properties of MreB correlate with changes in IDD.

To identify the MreB parameters that were most predictive of changes in cylindrical uniformity we performed a LASSO (Least Absolute Shrinkage and Selection Operator) regression[27]. LASSO is a machine learning method that involves penalizing the absolute size of the regression coefficients. The result is the smallest model within one standard error of the mean of the minimum LASSO regression. Because we did not know a priori whether measurements should be normalized per cell, per volume, or per surface area, we used all three normalizations as inputs into the LASSO regression. Combining all of our data, our LASSO analysis resulted in a model with four non-zero terms. These four terms included measures of polymer number, length, and curvature localization with different normalizations. However, a leave-one-out analysis of the data revealed that this was an over fit model (Supplementary Table 2). To determine which version of normalization was most predictive across different subsampled datasets, we used two different analysis methods (see Methods). Both methods converged on the same terms: MreB enrichment in regions of low Gaussian curvature (<2 μm$^{-2}$), the total length of MreB polymer in each cell normalized by cell volume, and the number of polymers per cell (Fig. 4, Supplementary Table 2). The combination of these three parameters was highly predictive of the change in cell shape ($r^2 = 0.93$), significantly more than any one parameter alone ($r^2 = 0.49$ (total polymer length), 0.68 (polymer number), and 0.52 (curvature localization)) (Fig. 4a, b). Importantly, this correlation holds for strain comparisons that have a positive or negative ΔIDD due to truncations in RodZ or MreB point mutations (Fig. 4a).

We next sought to experimentally test the LASSO regression's result, that changing specific MreB properties, like polymer number, should result in a predictable change in IDD. Because MreB$_{E143A}$ is able to maintain geometrically-localized MreB even in the absence of rodZ, this particular mutant enabled us to test our hypothesis. Ectopic expression of MreB$_{E143A}$ in a MreB$_{E143A}$ ΔrodZ background increased MreB polymer number (Supplementary Fig. 10). Importantly, we observed the LASSO-predicted change in shape upon increasing MreB polymer number, as ectopic expression of MreB$_{E143}$ made the cells more rod-like, even though rodZ was still absent (Fig. 4a; Table 1, Supplementary Fig. 10). We also ectopically expressed MreB$_{WT}$ in a MreB$_{WT}$ ΔrodZ background, which is not properly curvature-localized. This strain did not restore rod shape, confirming that the MreB$_{E143}$ effect is not a generic consequence of ectopic expression (Supplementary Fig. 10). These results support our conclusion that MreB-dependent uniform rod shape can be predicted by the presence of multiple polymers that are geometrically-localized and collectively long.

## Discussion

Our findings demonstrate that RodZ plays a central role in regulating both MreB's localization to curved subcellular regions and the number of MreB polymers per cell. Unbiased correlations between cell shape and MreB polymer properties in mreB and rodZ mutants further revealed that MreB curvature preference, along with MreB polymer number and length, can accurately predict E. coli cylindrical uniformity, while MreB curvature preference alone is insufficient to explain rod shape formation. Below we discuss the implications of RodZ's function as an MreB interaction partner. We also present a model in which distinct

aspects of MreB control distinct aspects of rod shape determination including rod initiation[2], centerline curvature[4,28] (modulated by MreB curvature localization), cell width determination (modulated by MreB angle)[5], and now, in this current work, cylindrical uniformity (modulated by multiple factors).

We previously showed that the transmembrane protein RodZ binds to MreB in the cytoplasm and cell wall synthesis enzymes in the periplasm. Both the complete loss of RodZ and truncations of RodZ reduced the motion of MreB, suggesting that RodZ helps couple MreB to cell wall insertion, which we hypothesized works through PBP2 and/or RodA[1]. In agreement with this hypothesis, RodA and PBP2 have recently been shown to exhibit MreB-like motion[29]. We also identified a strain, MreB$_{S14A}$ ΔrodZ, that significantly reduced MreB motion while retaining rod shape, suggesting that cell shape and rotation could be uncoupled. A more recent study demonstrated that there is still residual MreB rotation in MreB$_{S14A}$ ΔrodZ, but did not compare the motion of MreB$_{S14A}$ with and without RodZ[7]. The differences between our original study and this newer work may stem from the temperature-dependence of MreB rotation and differences in the data collection conditions. Nevertheless, even if RodZ is not absolutely necessary for MreB rotation in all conditions, the reduced MreB motion in the absence of RodZ suggests that RodZ normally promotes MreB rotation. While future studies will be needed to clarify how exactly MreB rotation is affected by both RodZ and temperature, here we demonstrate two additional functions for RodZ in regulating MreB curvature preference and polymer number.

First, we use 3D imaging to show that MreB localization is enriched near zero Gaussian curvature, which can explain why it preferentially localizes to the cylindrical region of the cell (whose Gaussian curvature is around zero) and avoid the cell poles (whose Gaussian curvature is positive). Oriented motion could promote MreB's geometric localization[7]. In vitro and in silico data indicate that MreB polymers have an intrinsic curvature, suggesting that MreB filaments could also potentially sense curvatures on their own[9,12,15]. However, our data show that in vivo, E. coli MreB$_{WT}$ polymers require RodZ to properly sense cell curvature. RodZ is not absolutely required for curvature localization as some MreB point mutants can localize to specific surface curvatures even in the absence of RodZ. Understanding the molecular mechanism by which RodZ influences MreB curvature localization will require in vitro systems that are currently unavailable. Since RodZ reaches around the MreB polymer and binds it on the side opposite the membrane[23], we speculate that the binding of RodZ could modulate the intrinsic curvature of MreB[10]. RodZ could also function to rigidify MreB polymers such that the absence of RodZ would cause MreB polymers to become more flexible, allowing them to bend more freely and therefore bind to a wider array of curvatures.

The second new role we discovered for RodZ is in regulating MreB polymer number. Previously, we had attributed changes in polymer number to changes in cell volume[5]. However, mutations in rodZ provided a shape-independent way to modulate polymer number, enabling us to conclude that polymer number promotes cylindrical uniformity independently of cell volume. There are several mechanisms by which RodZ could increase MreB polymer number. For example, RodZ could function as a nucleator that stimulates the formation of new polymers, as a severing protein that cuts single polymers into two separate polymers, as a capping factor that limits polymer growth, or as a factor that keeps polymers separated. We note that a simple polymer stabilization mechanism is unlikely because that would have led to significantly-increased polymer length. Regardless of the mechanistic details, whose dissection will require future in vitro studies, our findings represent the first identification of a factor that enhances MreB polymer formation.

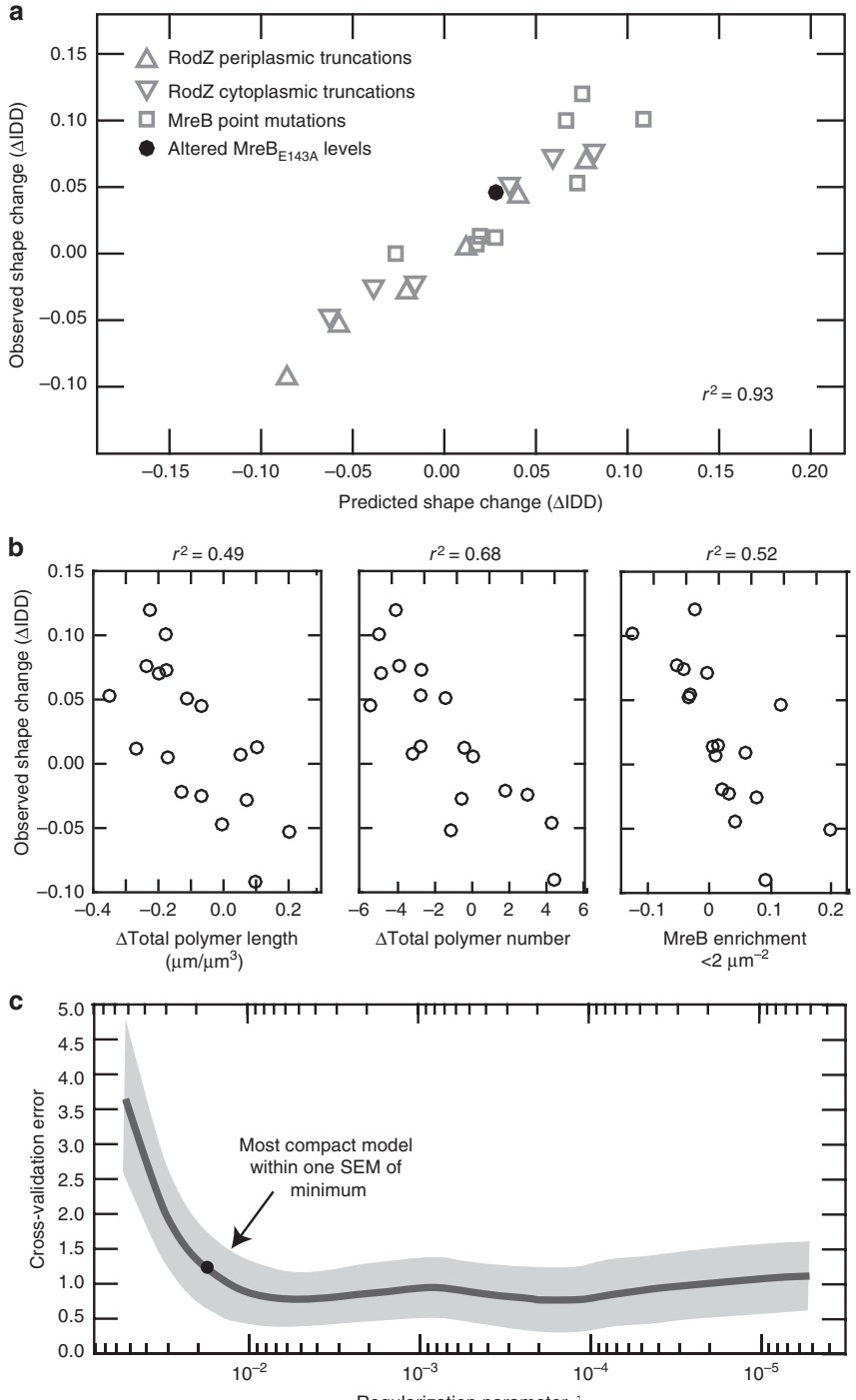

**Fig. 4** LASSO analysis reveals that rod shape requires many long and geometrically-localized MreB polymers. **a** The correlation between observed and predicted cell shape when using the LASSO model combining parameters. See Table 1 for all the strains used and the observed and predicted IDD values. Note that to preserve its use as a test of the model, overexpression of MreB$_{E143A}$ was not used for model selection and training. $r^2$ value represents the square of the Pearson correlation coefficient. **b** Left—the correlation between observed and predicted cell shape when only using polymer length normalized by volume. Middle—the correlation between observed and predicted cell shape when only using polymer number. Right—the correlation between observed and predicted cell shape when only using average MreB enrichment at Gaussian curvatures below 2 μm$^{-2}$. **c** The mean squared error (MSE) of 10-fold cross-validation as a function of the LASSO regularization parameter. The solid curve is the mean MSE and the shaded region represents one standard error of the mean. The dot represents the most compact model within one standard error of the mean from the minimum of the curve. See Supplementary Table 2 for the coefficients in this model

Interestingly, our RodZ truncation and MreB mutant analyses suggest that the functions of RodZ in promoting MreB rotation, polymer formation, and geometric localization are genetically separable. Thus, RodZ appears to use its cytoplasmic and periplasmic domains to promote multiple aspects of MreB, including

acting upstream of MreB assembly to increase polymer number and curvature localization and downstream of MreB assembly to enhance its coupling to the movement of the cell wall synthesis machinery. A role for RodZ in facilitating MreB's many functions could explain why specific point mutations in its multiple binding

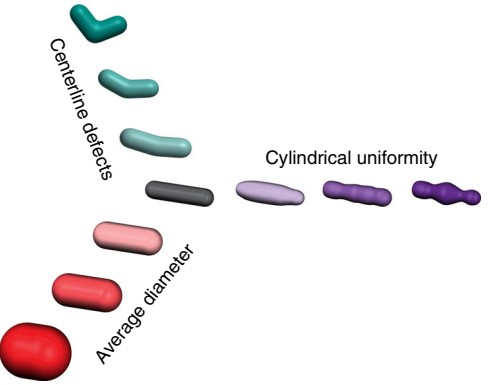

**Fig. 5** Simple, straight, rod-like cell shapes require multiple parameters to describe them. A straight rod is defined by its centerline curvature, cylindrical uniformity, and diameter. Deviations in any of these properties result in non-straight rods in extreme cases, and qualitatively ambiguous rods when only small changes occur. Teal—as centerline defects increase in magnitude cells become more bent until they are no longer straight. Purple —as cylindrical uniformity decreases cells exhibit increase fluctuations in their diameter along the long axis. Red—as width increases cells become more sphere like and less rod-like. For each of these shape descriptors, a quantitative metric of shape provides a continuous rather than a binary description of rod vs. non-rod

partners can suppress specific aspects of RodZ function[1,24,30,31]. Such modularity is an appealing way for the cell to tune its growth machinery in response to different growth conditions.

In previous studies, our lab and others determined that MreB promotes multiple aspects of rod shape determination[1,2,4,5]. MreB localization helps straighten rods and initiate new rods out of spheres, while MreB angle is correlated with the average width of a rod[2,5], even in the absence of RodZ (Supplementary Fig. 11). Here we show that MreB properties also correlate with the cylindrical uniformity of rods (Fig. 5). Specifically, a machine learning analysis (LASSO) of the correlations between MreB properties and rod shape revealed that a combination of modulating polymer number and total polymer length, along with curvature localization, is sufficient to accurately predict rod shape changes. While the LASSO analysis cannot distinguish whether MreB polymer number and total polymeric content (sum of the length of all polymers) directly impact cell shape or are correlates of other cellular processes, our result supports previous studies suggesting that MreB forms multiple independent structures distributed throughout the cylindrical portion of the cell[1,5,13,14,32,33].

The LASSO analysis determines the features that are most predictive of cylindrical uniformity and not the mechanism by which these features impact cell shape. Furthermore, the LASSO analysis incorporated all MreB structures, making no assumptions about which polymers are active. Consequently, the extremely high predictive power yielded by the LASSO analysis ($r^2$ = 0.93) was striking. This predictive power was confirmed by using the change in MreB polymer number induced by ectopic expression of $MreB_{E143A}$ in a $MreB_{E143A}$ $\Delta rodZ$ background to accurately predict the resulting change in cylindrical uniformity. Because both polymer number and total polymeric content are important, the LASSO results suggest that neither one long polymer nor a multitude of small polymers are sufficient to generate cylindrically uniform cells. We speculate that maintaining a rod shape with uniform diameter requires multiple MreB structures to make enough local shape measurements to direct the overall emergence of rod shape. In addition to promoting curvature sensing, long MreB structures could also distribute the area along which new cell wall material is being

inserted[4,17,34]. In the future, analysis of the effects of MreB and RodZ mutants on the pattern of cell wall insertion should help further elucidate the specific mechanisms by which MreB polymers affect cell wall synthesis and shape determination.

Our model for cylindrical uniformity predicts changes in cell shape ($\Delta$IDD) in a variety of backgrounds (mutations in both *mreB* and *rodZ*), and thus represents an additive model of cell shape. The fact that the aspects of MreB that predict cylindrical uniformity (polymer number, length, and curvature localization) are distinct from the aspect of MreB that predicts cell width (polymer angle) suggests that the absolute shape of the cell (width, straightness, uniformity, etc.) is a complicated function where different cell shape properties can be tuned independently (Fig. 5). Thus, while MreB is emerging as a central coordinator of rod shape, there must be additional factors important for determining specific aspects of morphology. Because RodZ influences both polymer number and MreB curvature localization, it will be important for future studies to unravel the specific contributions of curvature localization to the various aspects of rod shape formation.

## Methods

**Bacterial growth**. Bacteria were grown using standard laboratory conditions. Cultures were grown overnight in LB medium (10 g/L NaCl, 10 g/L tryptone, 5 g/L yeast extract), subcultured in the morning 1:1000, and grown to exponential phase (O.D.$_{600}$ 0.3–0.6) at 37 °C. Plasmids were electroporated into S17 *E. coli* and then subcloned into the appropriate strain using TSS transformation[35]. When required, 5 μM IPTG was used to induce gene expression. Antibiotics were used at the following concentrations: 100 μg/mL carbenicillin, 30 μg/mL chloramphenicol, 30 μg/mL kanamycin, and 10 μg/mL tetracycline. For a complete list of strains see Supplementary Table 1.

**Drug induced shape changes**. For mecillinam treatment, cells were grown overnight and subcultured in the morning 1:1000 into LB with sub-lethal concentrations of mecillinam (0.025–0.050 μg/mL). Cells were then grown to exponential phase (O.D.$_{600}$ 0.3–0.6) at 37 °C before being imaged, without washing or spinning, on agarose pads without mecillinam. For A22 treatment, cells were grown to early exponential phase (O.D.$_{600}$ 0.1–0.2) at 37 °C, and then treated at 37 °C for 1 h with A22 (1 or 20 μg/mL). Following treatment, cells were gently pelleted (5000 G for 60 s) and washed with fresh LB twice before being placed on agarose pads for imaging.

**Strain construction**. An MreB-GFP expressing strain deleted for *rodZ* was previously described[1]. This strain was used as the parent strain for the RodZ truncations. RodZ truncations were cloned into pTrc99A[36] and expressed without the addition of IPTG. MreB point mutants were previously isolated as A22 resistant mutants[5]. The *rodZ* deletion was moved into these strains via P1 transduction[37]. For a complete list of strains see Supplementary Table 1.

**Western analysis**. Whole-cell lysates were collected from exponentially grown cells. Approximately equal amounts of protein were loaded onto each gel and blots were treated with an anti-GFP antibody (from mouse, Roche 11814460001) to detect MreB-msfGFP$^{sw}$, and anti-σ$_{70}$ antibody as a loading control (from rabbit, a gift from the Silhavy Lab, Princeton University). The ratio of MreB to σ$_{70}$ was determined using a LI-Cor imager. To correct for differences between blots, a blotting control of WT MreB-msfGFP$^{sw}$ cells was included on each blot and the MreB/σ$_{70}$ ratio of each sample was normalized by the MreB/σ$_{70}$ control on that particular blot.

**Microscopy**. For all imaging, cells were grown at 37 °C in LB medium. Imaging was done on M63 Glucose plus caseamino acid pads with 1% agarose at room temperature.

Phase contrast images were collected on a Nikon90i epifluorescent microscope equipped with a 100×/1.4 NA objective (Nikon), Rolera XR cooled CCD camera (QImaging), and NIS Elements software (Nikon).

Images for 3D cell shape measurements were taken on a monolithic aluminum microscope (homemade) with a 100×/1.49 NA (Nikon) objective, iXon DU897 cooled EMCCD camera (Andor Technology), and a homemade LabView software package (National Instruments).

**Image acquisition**. For the 2D phase images data shown is from a representative experiment done in triplicate.

For 3D studies, a z-stack with a spacing of 150 nm was taken for the MreB channel. The number of z-stack images taken to determine the shape channel was between 45 and 55 depending on the size of the cells, 100 nm per step. The data represented is pooled from multiple days of image acquisition. Dividing cells were specifically excluded from any analysis.

**3D reconstructions**. 3D cell shape and polymer analysis utilized existing shape reconstruction software[1,5,38]. Briefly, to determine the 3D shapes of cells, z-stacks were taken of cells constitutively expressing cytoplasmic mCherry. We utilized a shape determination method that minimizes the difference between an observed z-stack and the forward convolution of a model with the experimental point spread function. For all strains, a set of triangular and a set of rectangular faces was used to parameterize the surface and measure the cell shape. These two reconstructions are similar to each other but represent the shape in a different framework. The triangular reconstruction is a set of faces each made up of three vertices ($S = \{F, V\}$) and the rectangular framework expands the surface as function of two one-dimensional "unwrapping" parameters, ($S(u, v)$). While the MreB intensity measurements are normalized by the surface area of each patch, the regularly sized triangular meshwork outperforms the rectangular framework in capturing the surface curvatures. This is particularly true at places like the poles of the cell which are singularities in the rectangular framework. The triangular framework was used to measure MreB curvature localization in all strains. This curvature enrichment was first normalized on a per-cell basis, meaning the average surface density in each cell was set to one (see Enrichment plot calculation). The rectangular framework is intrinsically a 2D surface and allowed us to reconstruct objects forced to live on the surface. We used this rectangular reconstruction framework to measure polymer statistics.

Polymers were measured by fitting the 3D MreB images to a set of polymers confined to the membrane. A 2D unwrapped image of the MreB polymers was created and used for intensity-based segmentation to determine the number of polymers and their location, length, and angle.

**Enrichment plot calculation**. To calculate the average concentration of MreB as a function of Gaussian curvature, we started with the ith cell, and calculated the total intensity of MreB fluorescence on the cell surface.

$$I_i^{\text{tot}} = \sum_j I_{i,j} \cdot A_{i,j} \tag{1}$$

The index $i$ is for each individual cell, the index $j$ is for each face in the 3D reconstructed shape, $I$ is the intensity of that face, and $A$ is the surface area of that face. This total intensity in cell $i$ was then converted into an estimate of the average density in that cell by dividing it by the total surface area.

$$I_i^* = \frac{I_i^{\text{tot}}}{\sum_j A_{i,j}} \tag{2}$$

Due to variations in illumination intensity and possible variations in the absolute abundance of MreB in each cell, each intensity value was normalized by $I^*$. At each surface element $i$, the Gaussian curvature $K$ was also evaluated. Each surface element contributes to the joint probability distribution of the intensity on the surface of each cell $P_i(I/I^*, K)$ weighted by its surface area. This joint probability was calculated by summing up all elements that came from that particular $(I/I^*, K)$ bin.

$$P_i\left(\frac{I}{I^*}, K\right) = \left(\sum_{j: \frac{I_{i,j}}{I_i^*} = \frac{I}{I^*}, K_{i,j} = K} A_{i,j}\right) \div \left(\sum_j A_{i,j}\right) \tag{3}$$

We then use Bayes' rule to calculate the conditional probability of finding a certain intensity at a particular curvature.

$$P_i(I/I^*|K) = \frac{P_i(I/I^*, K)}{P_i(K)} \tag{4}$$

The average intensity value at any particular curvature is then the expectation value of the intensity given this distribution.

$$\mathbb{E}[I/I^*|K]_i = \int P_i(I/I^*|K) \cdot I/I^* \cdot dI \tag{5}$$

We then average across all the cells for our estimate of the average MreB concentration at a particular curvature.

$$[\text{MreB}](K) = \langle \mathbb{E}[I/I^*|K]_i \rangle_{\text{cells}} \tag{6}$$

**Cell shape analysis**. Intracellular diameter deviation was modified from Morgenstein et al.[1] The diameter along the centerline was measured for each contour point and the standard deviations of these measurements were averaged within each cell (Supplementary Fig. 9). To account for differences in the magnitude of the standard deviation between different sized cells this average was divided by the mean diameter to form the coefficient of variation of the intracellular diameter deviation. We also made this measurement using 3D images of cells with triangular faces. We observed a high correlation between the 2D and 3D measurements (Supplementary Fig. 9).

SPACECRAFT was performed as previously reported[1] to determine differences between RodZ cytoplasmic truncation strains.

**MreB polymer length estimation**. We used two different approaches to estimate MreB polymer lengths. For the first approach, we calculated the average length of the measurable polymers (>200 nm) as reported in Fig. 3. Second, we wanted to include a value that represents an estimate for the average of all segmented MreB structures, including those under 200 nm. We observed that the distribution of polymer lengths appears to be a thresholded exponential, which can be explained by our inability to measure the lengths of polymers below a threshold of 200 nm (Supplementary Fig. 12). The average value of a thresholded exponential is mathematically defined as the threshold value plus the average value of the exponential. Therefore, the estimated average length of all segmented polymers is the average length of measured polymers (Fig. 3) minus the threshold length (200 nm). This predicts that roughly 60% of the all detected polymers should be >200 nm, consistent with our observations (Supplementary Data 1).

**LASSO regression**. LASSO regression is a type of linear least-squares regression that includes a regularization penalty ($\lambda$) for non-zero coefficients. This can be written as the set of parameters $\beta$ that minimizes

$$\frac{1}{N} \|y - \mathbf{X}\beta\|_2^2 - \lambda \|\beta\|_1, \tag{7}$$

where $N$ is the number of observations, $y$ is the observed cylindrical uniformity values, and $\mathbf{X}$ is a set of biophysical properties of MreB polymers (see Supplementary Table 2). Increasing $\lambda$ leads to selecting fewer non-zero coefficients at the cost of greater residual error.

For the regression of $\Delta$IDD against $\Delta\mathbf{X}_{\text{MreB}}$, there is a clear minimum in the mean squared error from cross-validation at around $\lambda$ to 0.02 (Fig. 4c), which we then chose as our regularization parameter for our further analysis. If two or more parameters are highly correlated, then the stochastic nature of the LASSO regression will randomly choose a parameter. In particular the various normalizations of parameters are highly correlated, for example polymer number/volume or area are highly correlated to each other. To see which version of normalization was most predictive across different subsampled datasets, we used two different analyses. In the first, we simply increased $\lambda$ further, to 0.022 and in the second we tested how well each combination of polymer number, polymer length and MreB enrichment performed in a 10-fold cross-validation test.

In k-fold cross-validation, the dataset is randomly split into $k$ subsets with approximately equal number of samples. The model is then trained against $k-1$ and asked to predict the observations in the final subset. The mean squared error from this testing is then saved as one value of the cross-validation. The training and testing process is then repeated using a different subset as the test set and the remaining $k-1$ used as training. The process is repeated until each subset has been used as the test set. This cross-validation protocol helps establish features and regression coefficients that are true predictors of the underlying phenomenon, not merely descriptors of noise in specific subsets.

While this cross-validation protocol led us to use the combination of total length/volume, polymer number/cell and enrichment $<2\,\mu\text{m}^{-2}$, our major conclusions that these are the three important characteristics that predict cylindrically uniform do not depend on the precise details of the normalizations. Not only does this combination perform better than all the other three parameter combinations, it also outperforms all three parameter combinations with two from these three and any one additional parameter (Supplementary Table 3).

**Code availability**. The custom MATLAB routines used for reconstructing 3D shapes utilizing active contours are freely available under a BSD 3-clause license and the latest version can be found at [https://github.com/PrincetonUniversity/shae-cellshape-public/]. This publication refers specifically to release 1.1.0 which has been archived at [https://doi.org/10.5281/zenodo.1256201].

**Data availability**. The data supporting the findings of the study are available in this article and its Supplementary Information Files. Supplementary Data 1 contains the summary data that were used as inputs into the LASSO regression. Additionally, the raw data that support the findings of this study are available from the corresponding author upon request.

# ARTICLE

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

## Acknowledgements

We thank Thomas Silhavy for the anti-σ70 antibody as well as Nicholas Martin and Robert Scheffler for assistance in Western blotting. We thank Matthias Koch and Elizabeth Ohneck for comments on the manuscript, the members of the Z.G. and J.W.S. laboratories for technical assistance and advice, and Ethan Garner for ongoing discussions on the interactions between MreB and RodZ. This work was supported by National Science Foundation PHY-1734030 (B.P.B., J.W.S.), the Glenn Centers for Aging Research (B.P.B.), National Institutes of Health R01 GM107384 (B.P.B., R.M.M., Z.G.), NIH R21 AI121828 (B.P.B., Z.G., J.W.S.), and NIH F32 GM103290 (R.M.M.). The funders had no role in study design, data collection and analysis, decision to publish, or preparation of the manuscript.

## Author contributions

B.P.B., J.W.S., Z.G., and R.M.M. conceptualized the project. B.P.B., J.W.S., Z.G., and R.M.M. designed the methodology. B.P.B. developed the software. B.P.B. and R.M.M. collected and analyzed the data; J.W.S. and Z.G. provided resources and project administration. B.P.B., J.W.S., Z.G., and R.M.M. wrote and edited the manuscript. B.P.B. and R.M.M. generated visualizations of the data.

## Additional information

**Competing interests:** The authors declare no competing interests.

