## [Peer Review File · Nature Communications]

Reviewers' comments:

Reviewer #1 (Remarks to the Author):

Since many years now, it has been known that the length, the orientation (relative to the short axis of the cell), and the motion of MreB filaments have a strong influence of rod shape cell morphology in many bacteria. It has also been shown that RodZ is very important for the function of MreB, and that RodZ directly binds to MreB. It has also been shown by several groups that point mutations in MreB have various effects in cell diameter, and can compensate for the lack of other key proteins involved in cell shape maintenance. In spite of all these data, it is still almost entirely unclear how MreB affects cell wall synthesis. Many models exist, but progress in the MreB field in recent years has been almost exclusively of descriptive nature, molecular mechanisms are lacking that would tell us how MreB works.

Morgenstein et al have performed several experiments elucidating the interplay between MreB and RodZ, in *Escherichia coli* cells. They find that RodZ affects the preference of MreB to localize to curved membranes, which to my knowledge is novel. They also show that the cytoplasmic domain of RodZ is essential for curvature sensing, which is not surprising because this is the part of RodZ that directly interacts with MreB. The authors show that "Cells need multiple, long, and geometrically-localized MreB polymers to grow as uniform rods". This has been shown in many studies since 2004. The discussion largely recapitulates points that have been shown before for MreB in different bacteria. Overall, the work is yet another descriptive study on localization properties of MreB, and of its interactor RodZ, which does not advance our knowledge on how MreB achieves its important function during cell growth.

Reviewer #2 (Remarks to the Author):

In this revision, the authors make strides towards improving the interpretability and rigor of the manuscript. In particular, I appreciate the authors' examination of the impact of cell volume on MreB polymer number, which supports their central conclusion that RodZ positively impacts MreB filament number. Importantly, the authors also clarified that the curvature enrichment analysis takes into account the available curvatures in each mutant, which was previously a source of concern.

Despite these improvements, I am still not convinced that RodZ directly regulates MreB curvature preference independent of its role as a nucleator. This is a central tenant of the manuscript and major source of potential impact. The authors attempted to quell this concern by treating WT cells with mecillinam, which inhibits PBP2 function and disrupts rod-shape, as a means to look at MreB localization independent of shape. However, based on the representative images shown in S1A and 1B, mecillinam treated wild type cells and rodZ null mutants do not adopt the same terminal morphology: the mecillinam treated cells shown still display some "rod-ness" while the rodZ null cells appear spherical. It remains possible these differences, if real, could be sufficient to confer MreB curvature enrichment observed for mec treated cells. To alleviate this concern, authors can include details on the morphological properties (i.e. aspect ratio, circularity, etc) on the cells included in this analysis to convince the readers that changes in morphology do not underlie the observed phenotype.

Reviewer #3 (Remarks to the Author):

The authors have satisfactorily addressed all my concerns. I support its publication with minor

revisions.

Line 473. Reference #1 does not have volume or page numbers.

Line 603. Please specify the statistical test was used to determine significance.

Line 770. "regression will randomly chose a parameter." *choose.

Line 801. Reference #2 does not have a volume or page numbers.

Line 850. Please specify error bars and statistical tests in supplementary figures.

Reviewer #4 (Remarks to the Author):

This work has a series of exciting findings, drawn from very solid observations. 1. Deletion of RodZ changes the curvature preference MreB and 2. RodZ controls the number of MreB polymers. Both of these points are well conducted, using satisfying quantification and careful attention to statistics. They also have satisfied a number of my questions. However, a few fundamental problems remain, ones that undermine a large fraction of the work. While the effects of RodZ on MreB are well done and solid, the rest of the paper focuses on the lengths, numbers, angles, and localization of MreB in an attempt to build a model for cell shape control. This is based upon an underlying, but unproven, assumption that draws their final conclusions and model for cell shape into question.

1. The primary issue with the paper is the underlying assumption that, in all their mutant conditions, the localization of MreB reflects sites of cell wall growth (or that this localization affects cell wall growth in any way). While this has been well demonstrated in Ursell 2014 for wild type cells, this becomes highly suspect in their RodZ knockouts, as in their last paper they showed that deleting RodZ stops MreB rotation, a hallmark of PG building activity. As RodZ is the factor that couples MreB to the rest of cell wall machinery, if it is removed, it cannot be assumed that MreB localization relates to the sites of cell wall insertion, or moreover, has anything to do with cell shape. This is further manifested by their conclusion that ""WT-like MreB localization is not sufficient for proper cell shape" - even though MreB is localized to the correct curvature (via their MreB mutants), the RodZ defective cells are not growing correctly.

While the authors may, as evidence, point to the fact they have mutations in MreB that allow cells to as rods, these mutations are completely uncharacterized, and their mechanism of shape suppression is unknown. This suppression could be direct (allowing cell wall synthesis to continue at MreB), or they could be indirect (somehow allowing cells to grow as rods with the enzymes building rod shape, uncoupled from MreB's location). If they want to correlate MreB localization to cell shape in this range of mutant combinations they need to show that, in the absence of RodZ, the mutant MreBs represent sites of growth. If not, then the suppression of Rod shape defects is indirect, and not reliant on MreB localization.

With this conservation, they machine learning model becomes questionable, as much of the data is from RodZ mutants. The machine learning cannot be used as proof their model is correct, as machine learning or modeling can predict any behavior, even if the inputs or assumptions are incorrect.

2. The authors use menicillium treatment as a way to generate round cells. This is treated as an independent perturbation - that menicillium only effect is to generate round cells. This is not a clean

experiment where the results can be so simply interpreted. B-lactams do far more than make cells round, they also activate multiple stress responses, create futile cycles, overall, throwing the cells into a perturbed state. It would not be surprising that this could affect or MreB filaments, detach them from the PBP complexes, or affect MreB localization. In summary, the conclusion - "These data show that RodZ specifically promotes MreB's curvature localization in a manner that is not merely secondary to its role in cell shape determination" is not solid. I'm sure they have cleaner experiments than I can devise, but perhaps they could make cells round with A22, then wash out the drug, and let MreB filaments reform, then observe MreB localization.

3. The claim that S14A or Y183N restores Wild Type MreB curvature location is an overstatement. "the curvature enrichment profile of MreBS14A was restored to a WT-like profile". Yes, both wild type MreB and the MreB mutants decrease at positive curvatures, but the MreB curvature profiles peak at zero, then decrease at negative values. The mutants do not decrease, and therefore are not the same as WT profiles.

4. Finally, curvature. The authors spend a lot of time in the response justifying their previous findings and old findings are the same, and that all data, findings, and models are the consistent. They do not need to justify this to this reviewer. rotation They need to be clear to the field, so that there is a referenceable, clear understanding as to whether MreB preferentially localizes to negative Gaussian curvature, or not?

They respond - "Furthermore, we have never before reported the Gaussian curvature enrichment of MreB". Please understand that, from both Ursell, and quotes in later papers and reviews referencing Ursell, they do state directly that MreB localizes not to "negative mean curvature", but "Negative Gaussian curvature".

- Ouzounov 2016, referencing Ursell - "During elongation, we observed that MreB has a preference to localize to regions of small or even negative local Gaussian curvature."

- Chang and Huang, 2016 referencing Ursell - "In E. coli cells, the actin-like MreB cytoskeleton localizes preferentially to regions of negative Gaussian curvature".

- Ursell 2014 - "To examine whether our results from 1D cell contours translated to the 2D surface curvatures that more accurately describe the physical interaction between MreB and the cell surface, we developed an imaging technique to quantify the mean and Gaussian curvatures of individual cells, which collectively specify whether a point on the surface is part of an outward bulge, an inward indentation, or a bend in the cell. We then correlated these data with MreB fluorescence (SI Materials and Methods, Fig. 1E, and Fig. S6)."

- Ursell 2014 - Materials and Methods to image MreB in 3D - "3D Surface Curvature Measurements. Cells were grown to OD₆₂₀ ~ 0.3 Image stacks were taken by moving the objective in 100-nm steps over 40 steps. MreB fluorescence was imaged using MG1655 mreBmsfGFP. Correlation data between cell shape and MreB localization were collected from 104 cells grown in sinusoidal channels."

As shown above, the idea that MreB localizes to Negative Gaussian curvature has been stated by this group multiple times. I have considered this one of the seminal findings in the MreB field, as have others. This was one of the few "core principles" of MreB, and it gave a model for rod shaped growth, and they way I and my students understood MreB's function. But now, these new and improved 3D measurements of MreB do not see this. Thus, we need to readdress this idea and overall model for MreB function. This is important also for other groups, as a recent paper examined MreB localization in 2D, again observing a small accumulation at negative mean curvature, which now I don't know how to

interpret.

Given the new improved imaging methods developed by this group, and the new findings of MreB localization, it is important they state that MreB does not localize to negative Gaussian curvatures, and why other methods give the wrong result. They also need to be clear about the old model and the implications, so others don't base their research or modeling on this ground truth.

Currently, this is ignored, and I fear if published, will hurt their credibility. They state they have added a part in the discussion to address this, but I cannot find it. Instead, the revised text contains even less mention of the "old localization" to negative curvatures, and how the new findings differ from this.

5. line 90 - "elongated polymers can coordinate the activity of multiple cell-wall modulating enzymes to produce twisting cylindrical growth." This is stated in a way as it were a fact (actual experimental data), but this is a concept - arising from simulations. This must be made clear that this is a speculation based on simulation, not data. I may be wrong, but to my understanding, no group has measured how many enzymes are present per filament, or if this is the basis of twisting of bacteria.

6. line 91-92 - "the orientation of MreB polymers relative to the cell axis helps determine the average cell width of the population." It has been shown that MreB filament angle correlates with cell width, not the MreB filament angle determines cell width. Establishing such a causal relationship is nearly impossible, especially as all of the experiments backing this claim are perturbing MreB and cell width at the same time.

7. It would really help the reader to understand the basis of RodZ increasing polymer number, as one point is puzzling, there is only a small increase (5%) in MreB associated with the membrane, yet a nearly doubling of the number of polymers. A few sentences explaining how this could occur would help.

8. Given the last, very satisfying paper from this first author, I'm a bit confused they do not test or discuss an obvious linkage. Their last paper showed that RodZ deletion stopped MreB motion of WT and mutant MreBs. In this work, they find that their MreB mutants have the same curvature localization as wild type, but that deletion of RodZ affects the curvature localization of mutant and normal MreB. This begs an obvious question, does MreB curvature localization require, or arise from MreB motion?

Reviewers' comments:

Reviewer #1 (Remarks to the Author):

Since many years now, it has been known that the length, the orientation (relative to the short axis of the cell), and the motion of MreB filaments have a strong influence of rod shape cell morphology in many bacteria. It has also been shown that RodZ is very important for the function of MreB, and that RodZ directly binds to MreB. It has also been shown by several groups that point mutations in MreB have various effects in cell diameter, and can compensate for the lack of other key proteins involved in cell shape maintenance. In spite of all these data, it is still almost entirely unclear how MreB affects cell wall synthesis. Many models exist, but progress in the MreB field in recent years has been almost exclusively of descriptive nature, molecular mechanisms are lacking that would tell us how MreB works.

Morgenstein et al have performed several experiments elucidating the interplay between MreB and RodZ, in *Escherichia coli* cells. They find that RodZ affects the preference of MreB to localize to curved membranes, which to my knowledge is novel. They also show that the cytoplasmic domain of RodZ is essential for curvature sensing, which is not surprising because this is the part of RodZ that directly interacts with MreB. The authors show that "Cells need multiple, long, and geometrically-localized MreB polymers to grow as uniform rods". This has been shown in many studies since 2004.

The discussion largely recapitulates points that have been shown before for MreB in different bacteria. Overall, the work is yet another descriptive study on localization properties of MreB, and of its interactor RodZ, which does not advance our knowledge on how MreB achieves its important function during cell growth.

Respectfully, we strongly disagree with the reviewer's assessment that it has been shown in multiple studies since 2004 that cells need multiple, long, and geometrically-localized MreB to form rods. These types of assumptions about the important features of MreB for its function that are based on intuitive assumptions rather than data. Until 2011 such assumptions led the field to think that MreB forms a single continuous structure. After that, seeing that the filaments rotate led the field to assume that the rotation is necessary for shape determination until we showed that rotation and rod shape could be uncoupled (Morgenstein 2015). Similarly, the fact that MreB prefers specific curvatures was first reported by our group in 2014, so the idea that this would have been self-evident a decade earlier makes little sense. What was really known before our current paper is that MreB has many properties: there are multiple filaments per cell, they have characteristic lengths, they localize to curvature, they have certain orientations, they associate with the membrane, etc. This current paper is the first in any bacterial species to take all of these and use an unbiased machine-learning approach to determine which of them are predictive specifically for cylindrical uniformity. We find the non-obvious result that a specific subset of MreB features (polymer number, total length, and curvature enrichment) are highly predictive of cylindrical uniformity. The fact that a different feature, polymer orientation, is not predictive of cylindrical uniformity but IS predictive of diameter (Ouzounov 2016) underscores the non-obvious nature of this finding and the importance of maintaining an unbiased perspective on MreB function.

Reviewer #2 (Remarks to the Author):

In this revision, the authors make strides towards improving the interpretability and rigor of the manuscript. In particular, I appreciate the authors' examination of the impact of cell volume on MreB polymer number, which supports their central conclusion that RodZ positively impacts

MreB filament number. Importantly, the authors also clarified that the curvature enrichment analysis takes into account the available curvatures in each mutant, which was previously a source of concern.

Despite these improvements, I am still not convinced that RodZ directly regulates MreB curvature preference independent of its role as a nucleator. This is a central tenant of the manuscript and major source of potential impact. The authors attempted to quell this concern by treating WT cells with mecillinam, which inhibits PBP2 function and disrupts rod-shape, as a means to look at MreB localization independent of shape. However, based on the representative images shown in S1A and 1B, mecillinam treated wild type cells and rodZ null mutants do not adopt the same terminal morphology: the mecillinam treated cells shown still display some “rod-ness” while the rodZ null cells appear spherical. It remains possible these differences, if real, could be sufficient to confer MreB curvature enrichment observed for mec treated cells. To alleviate this concern, authors can include details on the morphological properties (i.e. aspect ratio, circularity, etc) on the cells included in this analysis to convince the readers that changes in morphology do not underlie the observed phenotype.

We thank the reviewer for pushing us to clarify the important point of the shape-independence of MreB curvature enrichment yet further. We have added two things to reinforce our previous conclusions.

First, we have taken the reviewer’s suggestion and included our metric of rod-shape (IDD) for the population of mecillinam-treated cells, where we show that the mecillinam treatment disrupts cell shape to a similar extent as loss of RodZ (new Figure S1). The fact that $\Delta rodZ$ and mecillinam-treated cells can have such similar IDD’s and such different MreB curvature enrichment profiles indicates the independence of these measurements.

Second, following a suggestion from Reviewer 4, we have performed a second experiment to disrupt cell shape. Specifically, we treated cells with A22 and confirmed that cell shape was similarly affected to $\Delta rodZ$ and mecillinam-treated cells. A22 treatment disrupts MreB polymers such that we could not directly measure MreB curvature enrichment in these cells. However, upon washing out A22, the recovery of MreB polymer assembly is fast (minute time-scale) while the recovery of cell shape is slow (hour time-scale), so we washed out A22 from the rounded cells and examined MreB curvature enrichment quickly, before the cells recovered rod shape. Excitingly, we found that MreB was still enriched at low to negative Gaussian curvature values in these cells and depleted from higher positive Gaussian curvature values. If anything, the curvature enrichment was more pronounced in these cells (new Figure S3).

Reviewer #3 (Remarks to the Author):

The authors have satisfactorily addressed all my concerns. I support its publication with minor revisions.

Line 473. Reference #1 does not have volume or page numbers.

Line 603. Please specify the statistical test was used to determine significance.

Line 770. “regression will randomly chose a parameter.” *choose.

Line 801. Reference #2 does not have a volume or page numbers.

Line 850. Please specify error bars and statistical tests in supplementary figures.

We thank the reviewer for his/her support and have made all of the suggested edits.

Reviewer #4 (Remarks to the Author):

This work has a series of exciting findings, drawn from very solid observations. 1. Deletion of RodZ changes the curvature preference MreB and 2. RodZ controls the number of MreB polymers. Both of these points are well conducted, using satisfying quantification and careful attention to statistics. They also have satisfied a number of my questions. However, a few fundamental problems remain, ones that undermine a large fraction of the work. While the effects of RodZ on MreB are well done and solid, the rest of the paper focuses on the lengths, numbers, angles, and localization of MreB in an attempt to build a model for cell shape control. This is based upon an underlying, but unproven, assumption that draws their final conclusions and model for cell shape into question.

We thank the reviewer for acknowledging the improvement in the paper, but their comment that the second half of the paper “is based upon an underlying, but unproven, assumption that draws their final conclusions and model for cell shape into question” reveals a fundamental misunderstanding that we will attempt to clarify both in our responses below and in our revised text. As we will detail below, our approach is unbiased and makes no assumptions. We simply use machine learning to ask which features of MreB best PREDICT changes in IDD (cylindrical uniformity). Our final conclusions are simply that: there are specific parameters of MreB (namely polymer number, total polymer length, and curvature preference) that can accurately predict the cylindrical uniformity (smoothness) of a rod cell. In the discussion we develop a hypothesis to explain how and why this prediction holds true. But that’s not a conclusion – it’s a working model that we present as a helpful framework for guiding future experiments. Thus, the new insight in our paper is that specific properties of MreB can predict cylindrical uniformity and that those are different properties than those predict other cell shape features like cell width, indicating that MreB has multiple features that independently control distinct aspects of cell shape. We believe that it is unreasonable to ask us to prove a working model we present in the Discussion, especially since we presented that model with no objection from any of the reviewers in the previous round of review.

To address the reviewer’s concerns we have revised the text in the paper to make it clearer that the result that MreB polymer length/number/localization predict IDD holds independent of any assumptions whatsoever about the mechanism by which MreB functions (lines 320-323 and 465-468). Furthermore, we have clarified that the model we present is a working model and pointed out future ways to test the model such as determining whether MreB always represents sites of new growth (Lines 476-478).

1. The primary issue with the paper is the underlying assumption that, in all their mutant conditions, the localization of MreB reflects sites of cell wall growth (or that this localization affects cell wall growth in any way). While this has been well demonstrated in Ursell 2014 for wild type cells, this becomes highly suspect in their RodZ knockouts, as in their last paper they showed that deleting RodZ stops MreB rotation, a hallmark of PG building activity. As RodZ is the factor that couples MreB to the rest of cell wall machinery, if it is removed, it cannot be

assumed that MreB localization relates to the sites of cell wall insertion, or moreover, has anything to do with cell shape. This is further manifested by their conclusion that "WT-like MreB localization is not sufficient for proper cell shape" - even though MreB is localized to the correct curvature (via their MreB mutants), the RodZ defective cells are not growing correctly.

While the authors may, as evidence, point to the fact they have mutations in MreB that allow cells to grow as rods, these mutations are completely uncharacterized, and their mechanism of shape suppression is unknown. This suppression could be direct (allowing cell wall synthesis to continue at MreB), or they could be indirect (somehow allowing cells to grow as rods with the enzymes building rod shape, uncoupled from MreB's location). If they want to correlate MreB localization to cell shape in this range of mutant combinations they need to show that, in the absence of RodZ, the mutant MreBs represent sites of growth. If not, then the suppression of Rod shape defects is indirect, and not reliant on MreB localization.

As discussed above, our conclusions do not in fact rely on the assumption that MreB represents sites of growth in the absence of RodZ. Our conclusion is that MreB polymer number, total polymer length, and curvature enrichment are dependent on RodZ and predict cylindrical uniformity. This conclusion would be the same regardless of whether MreB still colocalizes with the sites of cell wall insertion in the absence of RodZ. In addition to not being necessary to support our current conclusions, showing that MreB still correlates with the sites of cell wall insertion would not actually prove our working model since it would not provide mechanistic insight into how the parameters we identified as important actually affect morphogenesis. In addition, the reviewer's request is unreasonable from a technical perspective. No group has ever reported colocalization of a protein and cell wall dynamics in 3D (which we require for our measurements) and the best available reagent to label walls, FDAAs, are not commercially available. So in summary, we agree with the reviewer that it would be nice to label cell wall insertion and MreB simultaneously and have proposed this as a future direction in the Discussion. However, we do not currently know if it's even possible to do this in 3D (it's never been done) and developing the new method, repeating it on all the strains we study and the paper is a massive undertaking that would be a paper on its own. Finally, even if we were to do this mountain of work, it's not clear that a positive result would significantly advance the paper (we'd be left with the same model and it would still be just a working model even if everything worked), and the paper would not be significantly affected if the result was negative (we'd still conclude that MreB polymer number/length/localization determine cylindrical uniformity independently of how MreB dictates cell width).

With this conservation, the machine learning model becomes questionable, as much of the data is from RodZ mutants. The machine learning cannot be used as proof their model is correct, as machine learning or modeling can predict any behavior, even if the inputs or assumptions are incorrect.

Again, we must reiterate that MreB-localized cell growth is not an input into the LASSO prediction and is thus not an assumption for its conclusions. We therefore do not see how this point is relevant. Our correlations are between measurable MreB characteristics like polymer number, length, etc., and measurable cell shape changes. MreB's localization to cell wall

synthesis is therefore not required for our predictive model. The machine learning is not “used as proof their model is correct” – it is merely used to conclude which parameters of MreB predict cylindrical uniformity. We tested this conclusion by determining whether modulating one of the resulting parameters had the predicted effect on cell shape, and it did. This proves that our machine learning is valid. The reviewer is correct that this does not prove any specific model of how those parameters affect shape, but we never make such a claim.

2. The authors use menicillium treatment as a way to generate round cells. This is treated as an independent perturbation - that menicillium only effect is to generate round cells. This is not a clean experiment where the results can be so simply interpreted. B-lactams do far more than make cells round, they also activate multiple stress responses, create futile cycles, overall, throwing the cells into a perturbed state. It would not be surprising that this could affect or MreB filaments, detach them from the PBP complexes, or affect MreB localization. In summary, the conclusion - "These data show that RodZ specifically promotes MreB's curvature localization in a manner that is not merely secondary to its role in cell shape determination" is not solid. I'm sure they have cleaner experiments than I can devise, but perhaps they could make cells round with A22, then wash out the drug, and let MreB filaments reform, then observe MreB localization.

We have followed this good suggestion from the reviewer: We first treated cells with A22 and confirmed that cell shape was similarly affected to $\Delta rodZ$ and mecillinam-treated cells (new Figure S1). A22 treatment disrupts MreB polymers, however, upon washing out A22, the recovery of MreB polymer assembly is fast (minute time-scale) while the recovery of cell shape is slow (hour time-scale), so we washed out A22 from the rounded cells and examined MreB curvature enrichment quickly, before the cells recovered rod shape. Excitingly, we found that MreB was still enriched at low to negative Gaussian curvature values in these cells and depleted from higher positive Gaussian curvature values. If anything, the curvature enrichment was more pronounced in these cells (new Figure S3).

3. The claim that S14A or Y183N restores Wild Type MreB curvature location is an overstatement. "the curvature enrichment profile of MreBS14A was restored to a WT-like profile". Yes, both wild type MreB and the MreB mutants decrease at positive curvatures, but the MreB curvature profiles peak at zero, then decrease at negative values. The mutants do not decrease, and therefore are not the same as WT profiles.

We have reworded this statement on lines 281-284.

4. Finally, curvature. The authors spend a lot of time in the response justifying their previous findings and old findings are the same, and that all data, findings, and models are the consistent. They do not need to justify this to this reviewer. rotation They need to be clear to the field, so that there is a referenceable, clear understanding as to whether MreB preferentially localizes to negative Gaussian curvature, or not?

They respond - "Furthermore, we have never before reported the Gaussian curvature enrichment of MreB". Please understand that, from both Ursell, and quotes in later papers and reviews referencing Ursell, they do state directly that MreB localizes not to "negative mean

curvature", but "Negative Gaussian curvature".

- Ouzounov 2016, referencing Ursell - "During elongation, we observed that MreB has a preference to localize to regions of small or even negative local Gaussian curvature."

- Chang and Huang, 2016 referencing Ursell - "In E. coli cells, the actin-like MreB cytoskeleton localizes preferentially to regions of negative Gaussian curvature".

- Ursell 2014 - "To examine whether our results from 1D cell contours translated to the 2D surface curvatures that more accurately describe the physical interaction between MreB and the cell surface, we developed an imaging technique to quantify the mean and Gaussian curvatures of individual cells, which collectively specify whether a point on the surface is part of an outward bulge, an inward indentation, or a bend in the cell. We then correlated these data with MreB fluorescence (SI Materials and Methods, Fig. 1E, and Fig. S6)."

- Ursell 2014 - Materials and Methods to image MreB in 3D - "3D Surface Curvature Measurements. Cells were grown to OD₆₂₀ ~ 0.3 Image stacks were taken by moving the objective in 100-nm steps over 40 steps. MreB fluorescence was imaged using MG1655 mreBmsfGFP. Correlation data between cell shape and MreB localization were collected from 104 cells grown in sinusoidal channels. "

As shown above, the idea that MreB localizes to Negative Gaussian curvature has been stated by this group multiple times. I have considered this one of the seminal findings in the MreB field, as have others. This was one of the few "core principles" of MreB, and it gave a model for rod shaped growth, and they way I and my students understood MreB's function. But now, these new and improved 3D measurements of MreB do not see this. Thus, we need to readdress this idea and overall model for MreB function. This is important also for other groups, as a recent paper examined MreB localization in 2D, again observing a small accumulation at negative mean curvature, which now I don't know how to interpret.

Given the new improved imaging methods developed by this group, and the new findings of MreB localization, it is important they state that MreB does not localize to negative Gaussian curvatures, and why other methods give the wrong result. They also need to be clear about the old model and the implications, so others don't base their research or modeling on this ground truth.

Currently, this is ignored, and I fear if published, will hurt their credibility. They state they have added a part in the discussion to address this, but I cannot find it. Instead, the revised text contains even less mention of the "old localization" to negative curvatures, and how the new findings differ from this.

We thank the reviewer for pushing us to further clarify the curvature enrichment of MreB. We think that the source of the confusion has to do with interpreting the curvature enrichment plots. While the eye may be drawn to a potential peak in MreB enrichment, all values above 1 are actually enriched and all values below 1 are depleted. We ask the reviewer to note that in our plots, the MreB enrichment is above 1 at all negative Gaussian curvature values (in a few cases they may approach 1, but they never dip below 1 to a statistically significant degree). Thus, our current paper shows that MreB is indeed enriched at negative Gaussian curvature, which is in agreement with the previous quotes the reviewer lists.

What is new in our current paper is that we previously had not published the full MreB enrichment curve for Gaussian curvature and merely described it. The analysis we present in

the new paper shows that while MreB is indeed enriched at negative Gaussian curvatures, its enrichment also extends to 0 and into low positive Gaussian curvature values (~2).

Consequently, to reflect this profile more accurately, in the current paper we have described MreB's enrichment as to "low-to-negative Gaussian curvatures". If the reviewer has a better suggestion for terminology we would welcome any suggestions.

Finally, the reviewer may be focusing on the presence of a peak in the enrichment around 0 Gaussian curvature. However, since MreB is still enriched (above 1) at negative values, for our claims it does not actually make a substantial difference where the MreB profile is highest or even if there is a peak at all.

Since we agree with the reviewer that this may be confusing we have done two things to try to make things clearer: 1) we added to the enrichment curves labels that make it clear that enrichment values over 1 are enriched, even if they are below the peak; 2) we added a brief description of our findings in an effort to clarify the curvature interpretation in the text.

5. line 90 - "elongated polymers can coordinate the activity of multiple cell-wall modulating enzymes to produce twisting cylindrical growth." This is stated in a way as it were a fact (actual experimental data), but this is a concept - arising from simulations. This must be made clear that this is a speculation based on simulation, not data. I may be wrong, but to my understanding, no group has measured how many enzymes are present per filament, or if this is the basis of twisting of bacteria.

We have reworded this on line 90.

6. line 91-92 - "the orientation of MreB polymers relative to the cell axis helps determine the average cell width of the population." It has been shown that MreB filament angle correlates with cell width, not the MreB filament angle determines cell width. Establishing such a causal relationship is nearly impossible, especially as all of the experiments backing this claim are perturbing MreB and cell width at the same time.

We have reworded this on line 103-104.

7. It would really help the reader to understand the basis of RodZ increasing polymer number, as one point is puzzling, there is only a small increase (5%) in MreB associated with the membrane, yet a nearly doubling of the number of polymers. A few sentences explaining how this could occur would help.

The reviewer raises an interesting observation. There are a large number of explanations for how RodZ might affect MreB polymer without affecting the fraction of membrane associated MreB. For example: 1) MreB monomers may also be membrane associated, 2) RodZ could similarly affect MreB membrane association and dissociation, 3) RodZ could act as a bundling inhibitor such that in the absence of RodZ we see fewer polymer structures because polymers have collapsed onto one another. We have added this final model, which is mechanistically distinct than what we had originally proposed into the Discussion on line 499-500 and thank the reviewer for pushing us to consider additional mechanisms for RodZ function.

8. Given the last, very satisfying paper from this first author, I'm a bit confused they do not test or discuss an obvious linkage. Their last paper showed that RodZ deletion stopped MreB motion of WT and mutant MreBs. In this work, they find that their MreB mutants have the same curvature localization as wild type, but that deletion of RodZ affects the curvature localization of mutant and normal MreB. This begs an obvious question, does MreB curvature localization require, or arise from MreB motion?

We have shown that MreBS14ArodZ disrupts MreB rotation but retains wt-like curvature enrichment, such that MreB localization does not require or result from MreB rotation.

Reviewers' comments:

Reviewer #2 (Remarks to the Author):

The authors have sufficiently addressed my previous concerns.

However, in the time since the manuscript has been last reviewed, work by Hussain et. al (2018) challenges the authors' previous findings (Morgenstein 2015) that RodZ mediates directional motion of MreB and motion can be decoupled from MreB's influence on cell shape. In fact, the Hussain and colleagues used the same strain and imaging conditions used in Morgenstein 2015 and note the presence of processive directional motion of MreB across cells and imaging conditions independent of the presence of RodZ. The authors state this finding was further validated by an independent group and even by the Gitai group themselves. Hussain et al. summarize their findings as follows in the discussion of their paper, as well as in the author response to reviewer comments:

From discussion:

In contrast to our model, a previous study in *E. coli* concluded that directional MreB motion was not required for rod shape. This conclusion was based on the observation that cells lacking RodZ are still rod-shaped, even while they observed no directional motion of GFP-MreB(S14A) filaments (Morgenstein et al., 2015). We acquired this strain from the Gitai lab (Δ rodZ, mreBS14A-msfGFP), sequenced the mutations, and examined the motions of MreB with TIRFM. Surprisingly, we observed directional motions of mreBS14A-msfGFP filaments in almost every cell (Figure 7—video 2) and subsequent analysis indicated these movements were processive (Figure 7—figure supplement 1). We could observe directional motion across a range of growth conditions, although at lower temperatures, or in less rich media (the conditions used for imaging in Morgenstein et al), it appeared that smaller fraction of the mreBS14A-msfGFP moved directionally. Thus, in contrast to the conclusions of (Morgenstein et al., 2015): (1) the directional motion of MreB has not yet been uncoupled from rod shape, and (2) RodZ is not required for directional MreB motion. We suspect that the conclusions of Morgenstein et al. arose from growing the cells in rich media at high temperatures, then imaging the cells in less rich media at low temperatures, where only a small fraction of the MreB filaments move directionally.

Although this finding does not necessarily undercut the value of the present study, the authors do not do themselves a favor by ignoring the controversy of their previous finding. Thus, the text should be extensively amended as to not propagate misinformation about the roles of RodZ in MreB dynamics and cell shape maintenance. Examples of sections that should be amended include the following:

Line 37: "Here we show that in addition to its previously-described role in mediating MreB motion..."

Line 106-108: "We previously showed that RodZ functions downstream of MreB as an adaptor that causes MreB to rotate around the cell circumference..."

Line 111: "Here we show that in addition to its role in promoting MreB rotation..."

Line 124-127: "We recently showed that the transmembrane protein RodZ interacts with both MreB and the cell wall synthesis machinery to couple MreB rotation to cell wall synthesis¹. RodZ promotes MreB rotation and specific point mutations in..."

Reviewer #4 (Remarks to the Author):

After many rounds of review, the circular arguments, and brush over interpretations in this paper still remain. This paper is not publishable in its current form, and it appears the reviewers are too entranced by their model, which is based on speculative and likely incorrect assumptions, for this to ever achieve resolution.

So, I am going to agree with reviewer 1's last statement. The RodZ data is solid, and could be a paper by itself. The curvature localization data for the wild type MreB is pretty much the same as before, (or a slight change), and does not extend on the field.

But beyond that, the curvature localization data of the RodZ with their suppressor mutant, while it could be interesting, is problematic due to the fact that its activity (and therefore relevance to cell growth) cannot be proven, and now is even more suspect. Given that the model is based on this not only doubtful, but now highly suspect data, the model becomes speculative, and draws into serious doubt many of the conclusions about RodZ/MreBs effects on cell uniformity.

1. RodZ's effect on curvature localization based on mutants, and the effects on cell shape.

A main conclusion is "Rod-shaped cells have WT-like MreB localization" And In their response - "We have shown that MreBS14A rodZ disrupts MreB rotation but retains wt-like curvature enrichment, such that MreB localization does not require or result from MreB rotation."

A- Here is a fundamental problem with the paper. They lump the curvature localizations into "WT" and "Wt-Like". It is obvious there are 3 different curvature localizations. 1. The normal, (new) wt curves have a hump, peaking at 0. 2. Their mutant curves are linear, with large accumulations at negative, unlike wild type. 3. Flat. With few filaments. Thus, lumping the S14A and Y183N into "Wt-like" is no means correct, and a vast over simplification to shoehorn the data into their model. The curves of S14A and Y183N are in no way near wild type localization.

B - "We have shown that MreBS14A rodZ disrupts MreB rotation".

Given the recent collaboration showing this conclusion was incorrect, (due to unknown reasons). This is not true and a misrepresentation of this mutant.

2. At the base of this model is the unproven assumption that MreB localization given negative curvature leads to growth at that spot. Not only is that point still unproven (despite them stating it as proven repeatedly), it becomes a problem if they don't know If their mutant filaments are "active".

If the goal of model is to predict cell wall uniformity, it relies on their measures going into it to be reflective of sites of growth. However, a great deal of the data put into their model is from their various MreB suppressor mutants, for which they have no way of verifying if the localization of these mutants reflects cell wall synthesis. While this was a problem before, it is now even more so, given their recent collaboration reexamining this exact mutant, which suggests most of these filaments do not reflect sites of cell wall synthesis. In the absence of RodZ, the S14A mutant does move, but only a small bit of it, and moves at increased rates of growth. However, even then, it looks like a lot of it does not (unfortunately not quantitated). This indicates that the large amount of "not moving" S14A filaments are inactive, not sites of insertion. Thus, the use of the S14A mutant (as well as all the others) as either data in respect to its localization, or as points of input for their model for cell shape become highly suspect: 1) not only might the curvature localization of the moving and not moving S14A filaments be different, 2) If most S14A filaments don't move, all available evidence suggests these filaments are not associated with any cell wall synthesis.

Basically - if this mutant, or any of the others is not making cell wall, making a model of cell shape based on their localization cannot be justified. They may as well look at the localization of CTP synthesis filaments, or some other inert filament that does nothing to the cell wall growth. Thus, to use and interpret the effects of this mutant on localization and cell shape, they need to understand it. The same is true for what happens with the S14A mutant when rodZ is present. What fraction of those filaments are active? Is there a difference in their localization?

Therefore the claim "We find that curvature preference is necessary but not sufficient to grow cylindrically uniform cells, while a combination of MreB polymer number, total polymer length, and curvature preference accurately predict changes in cylindrical uniformity." Is based on a suppressor mutant whose localization, all available evidence suggests, has nothing to do with cell wall synthesis. Thus, this statement, the LASSO model they build, and their conclusions from it, are very lacking in their validity.

While the authors might respond "but MreB must be correctly localized for rod shape", they must note that is circular reasoning, and (as detailed below), a point that has not been proven: no group has been able to prove curvature localization causes, or simply just correlates with cell shape defects. The curvature localization could be simply a reflection of some other problem with the filaments or PBPs.

"We have shown that MreBS14ArodZ disrupts MreB rotation but retains wt-like curvature enrichment, such that MreB localization does not require or result from MreB rotation."

As above, this is an especially problematic statement, as given their recent collaboration, it undercuts their entire model. If MreB filaments are not moving, then they are not active. If the curvature localization of these "dead" filaments don't reflect cell wall insertion, how do they effect cell shape?

3. Finally, I would like to point out to both the editor and the authors that a large number of underlying assumptions in this work are stated both in the introduction and text as fact. But many of these points not validated, rather are models (or hypothesis), thus these claims are blatant over interpretations. When the authors make these claims as fact (and base their subsequent models on them) without them being verified, they undermine the both their own credibility and the readers trust in the paper.

Primarily – a fundamental problem underlying this paper is that it is based on two unproven points. No matter how many times this is stated as "fact", investigation of the literature shows these are unproven assumptions.

A- correct curvature localization is required for wild type cell shape.
(As mentioned in the response to reviewer 1.) "since 2004 that cells need multiple, long, and geometrically-localized MreB to form rods."

While MreB might have a slight accumulation at negative curvature, no group, this one or any other, has shown curvature localization is causative of shape: MreB mutants have perturbed shapes, and perturbed curvature localizations. All of these points correlate, but they have not been uncoupled, perturbed curvature localization has not been shown to be causative. These mutants could be affecting many other properties of the filaments or its interactions, resulting in these shapes. (to claim this, one would have to make filaments localize to the wrong spot without affecting the filaments or any of the cell wall synthesis proteins).

B- MreB curvature localization reflects increased sites of growth.

Lines 69-72 "In E. coli, the bacterial actin homolog MreB organizes cell wall insertion by localizing to regions of the cell with particular geometric curvatures and recruiting cell wall enzymes to direct growth to those sites"

This is completely false. While MreB does show a very slight (2%?) accumulation at given curvatures, no work has shown either 1) recruitment of cell wall enzymes to those site, Or More importantly 2), no work has shown that this accumulation leads to increased growth at those sites. Rather, recent work by the VanTeffleen group demonstrates curvature localization does not increase growth, a point not cited nor discussed.

Lines 41-43 "Quantitative 3D measurements and a series of mutant strains show that among various properties of MreB, polymer number, total length of MreB polymers, and MreB curvature preference are the key determinants of cylindrical uniformity"

This data (part of which is suspect) shows corrections to cellular uniformity, not the determination of cellular uniformity.

Lines 61-62 " Previously we studied the determinants of E. coli straightening and population-average width, but the determinants of single-cell cylindrical uniformity remain unclear"

Here they reference their previous paper, which showed that the angle of filaments correlates with width. Again, because they could not independently modulate width, and measure angle, not angle, and width, the properties correlate, but they do not "determine" .

Lines 87-88 First, elongated polymers may coordinate the activity of cell-wall modulating enzymes to produce the twisting cylindrical growth seen in E. coli.

I'm shocked they use this reference as evidence for this statement. This work, by Wang, showed E. coli twist matched the twist of spirals of a problematic MreB fusion, whose spiral localization (and angle) was an artifact. This exact point was corrected by this group in a later work, that demonstrated a much more shallow angle. Why would they reference the angle of an artifactual filament structure as evidence for chiral growth, when the angle of cell twist no longer matches the real angle of MreB filaments?

REVIEWER COMMENTS

REFEREE 2 (bacterial cell biology, cell division, FtsZ assembly)

The authors have sufficiently addressed my previous concerns.

We are glad that we were able to address the previous concerns of the reviewer.

However, in the time since the manuscript has been last reviewed, work by Hussain et. al (2018) challenges the authors' previous findings (Morgenstein 2015) that RodZ mediates directional motion of MreB and motion can be decoupled from MreB's influence on cell shape. In fact, the Hussain and colleagues used the same strain and imaging conditions used in Morgenstein 2015 and note the presence of processive directional motion of MreB across cells and imaging conditions independent of the presence of RodZ. The authors state this finding was further validated by an independent group and even by the Gitai group themselves. Hussain et al. summarize their findings as follows in the discussion of their paper, as well as in the author response to reviewer comments:

From discussion: *In contrast to our model, a previous study in E. coli concluded that directional MreB motion was not required for rod shape. This conclusion was based on the observation that cells lacking RodZ are still rod-shaped, even while they observed no directional motion of GFP-MreB(S14A) filaments (Morgenstein et al., 2015). We acquired this strain from the Gitai lab ($\Delta rodZ$, mreBS14A-msfGFP), sequenced the mutations, and examined the motions of MreB with TIRFM. Surprisingly, we observed directional motions of mreBS14A-msfGFP filaments in almost every cell (Figure 7—video 2) and subsequent analysis indicated these movements were processive (Figure 7—figure supplement 1). We could observe directional motion across a range of growth conditions, although at lower temperatures, or in less rich media (the conditions used for imaging in Morgenstein et al), it appeared that smaller fraction of the mreBS14A-msfGFP moved directionally. Thus, in contrast to the conclusions of (Morgenstein et al., 2015): (1) the directional motion of MreB has not yet been uncoupled from rod shape, and (2) RodZ is not required for directional MreB motion. We suspect that the conclusions of Morgenstein et al. arose from growing the cells in rich media at high temperatures, then imaging the cells in less rich media at low temperatures, where only a small fraction of the MreB filaments move directionally.*

Although this finding does not necessarily undercut the value of the present study, the authors do not do themselves a favor by ignoring the controversy of their previous finding. Thus, the text should be extensively amended as to not propagate misinformation about the roles of RodZ in MreB dynamics and cell shape maintenance. Examples of sections that should be amended include the following:

Line 37: "Here we show that in addition to its previously-described role in mediating MreB motion...."

Line 106-108: "We previously showed that RodZ functions downstream of MreB as an adaptor that causes MreB to rotate around the cell circumference..."

Line 111: "Here we show that in addition to its role in promoting MreB rotation..."

Line 124-127: "We recently showed that the transmembrane protein RodZ interacts with both MreB and the cell wall synthesis machinery to couple MreB rotation to cell wall synthesis¹. RodZ promotes MreB rotation and specific point mutations in..."

As a first point, we appreciate the reviewer pointing out that the discrepancy between the observations in Hussain et al. 2018 and Morgenstein et al. 2015 "does not necessarily undercut the value of the current study". Indeed we want to stress that the extent to which RodZ promotes MreB rotation has no bearing on whether RodZ also affects MreB polymer number and curvature localization and that our current work's findings are sound regardless. Nevertheless, since a similar point was brought up by reviewer 4, we agree that it is beneficial to clarify and underscore the independence of the current work from the role of RodZ in MreB motion and have thus included a new section in the discussion on this topic.

Second, the discrepancies between Hussain/Morgenstein are still not entirely resolved, and we thank the reviewer for pushing us to make our own wording more clear, which we have done in multiple places in the main text. While the discussion of the discrepancy is not particularly important to this manuscript, we would like to point out a distinction between the notion of RodZ's role as *required for rotation* as compared to *promoting MreB rotation*. In the data published by Hussain et al., the authors only show data for MreB_{S14A} Δ RodZ. This enables them to observe that there is still some residual processive MreB motion in the absence of RodZ and conclude that RodZ is not absolutely required for MreB motion. However, it does little to address the related question, "Does RodZ promote MreB's processive motion?" As we showed in Morgenstein 2015, deleting *rodZ* resulted in decreased processive motion in both the MreB_{WT} and MreB_{S14A} backgrounds. During our dialog with the Garner group over the past few months (some of which is included in the response to reviewers, Hussain et al. 2018), both we and the Garner group have come to two conclusions: (A) Δ RodZ MreB_{S14A} shows a reduced amount of MreB processive motion relative to MreB_{WT} and (B) Δ RodZ MreB_{S14A} still has some processive motion at 37°C, the amount of which requires further investigation. Furthermore, we have shown in multiple labs that in the conditions measured in Morgenstein et al.,

Δ RodZ MreB_{S14A} displays significantly reduced motion relative to MreB_{S14A} with RodZ (a strain not measured by Hussain). In the revised text we have taken care to revise all of the specific statements cited above to reflect that RodZ promotes MreB rotation and is not fully required for it. We have also included a specific discussion of the Hussain finding in the Discussion.

REFeree 4 (super-resolution microscopy, bacterial cell division, FtsZ assembly)

After many rounds of review, the circular arguments, and brush over interpretations in this paper still remain. This paper is not publishable in its current form, and it appears the reviewers are too entranced by their model, which is based on speculative and likely incorrect assumptions, for this to ever achieve resolution.

Many of reviewer 4's comments throughout the review process appear to stem from a major misconception about our machine learning approach. It may be helpful to clarify two different usages of the word "model". The first "model" is a type of mental hypothesis that describes how all the known data work together to describe the physical phenomenon. This could be described as a working model or a mental model. We are sympathetic to the reviewer's perspective that it is important to clarify what we really "know" about MreB/RodZ and what is just a speculative "working model". In revising the paper we have gone to great lengths to do just that: in both the intro and discussion we try to always state the specific results and then make it clear when we interpret them to suggest a working model. We thank the reviewer for pushing us in this direction as we feel that the revised text is clearer, more accurate, and better explains how this paper's findings extend previous findings. Importantly, however, none of the current paper's conclusions depend on any specific model of MreB function and thus remain sound.

The second type of "model" is a type of quantitative data analysis, where one builds a predictive, mathematical description for a set of observations. For example, one could model the fall of a sphere under the influence of gravity and predict the position of the ball as a function of time. It is this second type of "quantitative model" that the LASSO regression allows us to build. The LASSO machine learning framework asks the question, "which MreB parameters, when used together, best predict changes in cylindrical uniformity?" It makes no assumptions about whether those parameters are directly coupled to cell wall synthesis, or what mechanism brings about the correlation. In describing these results, we have attempted to differentiate where this predictive model ends and how it fits into our working model. We appreciate that reviewer 4 has pushed us to be clarify the points in our text where they find our language to be unclear. However, we cannot stress enough that our "working model" is in no way fundamental to or the basis for our "predictive model". Our paper reports two core findings: (A) RodZ affects MreB polymer number and curvature preference and (B) MreB polymer number, length, and curvature preference correlate with cylindrical uniformity in a predictive quantitative model. These findings are sound regardless of how/if MreB actually mediates cell shape regulation and are thus independent of any preferred "working model".

So, I am going to agree with reviewer 1's last statement. The RodZ data is solid, and could be a paper by itself. The curvature localization data for the wild type MreB is pretty much the same as before, (or a slight change), and does not extend on the field.

We thank the reviewer for supporting the publication of the RodZ data. We also agree with the reviewer that the localization data for wild-type MreB is "*pretty much the same as before*". These data were not meant to extend the field but rather served as controls for our discovery of a new role for RodZ in the localization of MreB.

But beyond that, the curvature localization data of the RodZ with their suppressor mutant, while it could be interesting, is problematic due to the fact that its activity (and therefore relevance to cell growth) cannot be proven, and now is even more suspect. Given that the model is based on this not only doubtful, but now highly suspect data, the model becomes speculative, and draws into serious doubt many of the conclusions about RodZ/MreBs effects on cell uniformity.

Please see the discussion above for clarification of the term "model" and its relevance to our conclusions. We have attempted to address the reviewer's specific objections below.

1. RodZ's effect on curvature localization based on mutants, and the effects on cell shape.

A main conclusion is "Rod-shaped cells have WT-like MreB localization" And In their response - "We have shown that MreBS14A rodZ disrupts MreB rotation but retains wt-like curvature enrichment, such that MreB localization does not require or result from MreB rotation."

A- Here is a fundamental problem with the paper. They lump the curvature localizations into "WT" and "Wt-Like". It is obvious there are 3 different curvature localizations. 1. The normal, (new) wt curves have a hump, peaking at 0. 2. Their mutant curves are linear, with large accumulations at negative, unlike wild type. 3. Flat. With few filaments.

Thus, lumping the S14A and Y183N into “Wt-like” is no means correct, and a vast over simplification to shoehorn the data into their model. The curves of S14A and Y183N are in no way near wild type localization.

The reviewer is correct that we lumped all of their class 1 and class 2 profiles together as they show *geometric enrichment*, which is distinct from MreB’s lack of geometric localization in Δ RodZ. Nevertheless, we concede that deciding which features of MreB curvature enrichment are “WT-like” is subjective. We have therefore completely eliminated the term “WT-like” from the revised text and clarified our wording to reflect which strains show geometric enrichment/localization.

B - “We have shown that MreBS14A rodZ disrupts MreB rotation”.

Given the recent collaboration showing this conclusion was incorrect, (due to unknown reasons). This is not true and a misrepresentation of this mutant.

Please see our response to Reviewer 2 above which clarifies this issue.

2. At the base of this model is the unproven assumption that MreB localization given negative curvature leads to growth at that spot. Not only is that point still unproven (despite them stating it as proven repeatedly), it becomes a problem if they don’t know if their mutant filaments are “active”.

If the goal of model is to predict cell wall uniformity, it relies on their measures going into it to be reflective of sites of growth. However, a great deal of the data put into their model is from their various MreB suppressor mutants, for which they have no way of verifying if the localization of these mutants reflects cell wall synthesis. While this was a problem before, it is now even more so, given their recent collaboration reexamining this exact mutant, which suggests most of these filaments do not reflect sites of cell wall synthesis. In the absence of RodZ, the S14A mutant does move, but only a small bit of it, and moves at increased rates of growth. However, even then, it looks like a lot of it does not (unfortunately not quantitated). This indicates that the large amount of “not moving” S14A filaments are inactive, not sites of insertion. Thus, the use of the S14A mutant (as well as all the others) as either data in respect to its localization, or as points of input for their model for cell shape become highly suspect: 1) not only might the curvature localization of the moving and not moving S14A filaments be different, 2) If most S14A filaments don’t move, all available evidence suggests these filaments are not associated with any cell wall synthesis.

Basically - if this mutant, or any of the others is not making cell wall, making a model of cell shape based on their localization cannot be justified. They may as well look at the localization of CTP synthesis filaments, or some other inert filament that does nothing to the cell wall growth. Thus, to use and interpret the effects of this mutant on localization and cell shape, they need to understand it. The same is true for what happens with the S14A mutant when rodZ is present. What fraction of those filaments are active? Is there a difference in their localization? Therefore the claim “We find that curvature preference is necessary but not sufficient to grow cylindrically uniform cells, while a combination of MreB polymer number, total polymer length, and curvature preference accurately predict changes in cylindrical uniformity.” Is based on a suppressor mutant whose localization, all available evidence suggests, has nothing to do with cell wall synthesis. Thus, this statement, the LASSO model they build, and their conclusions from it, are very lacking in their validity.

While the authors might respond “but MreB must be correctly localized for rod shape”, they must note that is circular reasoning, and (as detailed below), a point that has not been proven: no group has been able to prove curvature localization causes, or simply just correlates with cell shape defects. The curvature localization could be simply a reflection of some other problem with the filaments or PBPs.

“We have shown that MreBS14ArodZ disrupts MreB rotation but retains wt-like curvature enrichment, such that MreB localization does not require or result from MreB rotation.”

As above, this is an especially problematic statement, as given their recent collaboration, it undercuts their entire model. If MreB filaments are not moving, then they are not active. If the curvature localization of these “dead” filaments don’t reflect cell wall insertion, how do they effect cell shape?

We respectfully disagree with the reviewer. As we discuss above, whether MreB localization corresponds to sites of cell growth or whether a specific MreB filament is active is immaterial to our conclusion that the number, length, and curvature preference of MreB filaments can predict cylindrical uniformity. These arguments do impact “working models” for WHY these features of MreB are predictive, but we have clarified that our conclusion is merely that these features DO predict cylindrical uniformity.

The reviewer also focuses a great deal on the S14A mutant. The LASSO quantitative model incorporates all of the data we measured across all strains and makes no subjective assumptions about specific “active” or “inactive” populations (the reviewer is making his/her own assumption in asserting that only the moving filaments are active, which has never been shown). Our conclusions thus do not depend on any specific interpretation of how S14A affects MreB or RodZ. We merely conclude that across all the different mutants and truncations examined, three properties

of MreB (polymer number, length, and curvature preference) are sufficient to predict cylindrical uniformity with an R^2 of 0.93. We furthermore have included all of our data in Fig. 4 and coded them to allow the readers and reviewers to focus on specific classes. Even if one ignores all the square points (the data derived from MreB point mutants), one sees that the conclusions of the correlations remain the same.

3. Finally, I would like to point out to both the editor and the authors that a large number of underlying assumptions in this work are stated both in the introduction and text as fact. But many of these points not validated, rather are models (or hypothesis), thus these claims are blatant over interpretations. When the authors make these claims as fact (and base their subsequent models on them) without them being verified, they undermine the both their own credibility and the readers trust in the paper.

Primarily – a fundamental problem underlying this paper is that it is based on two unproven points. No matter how many times this is stated as “fact”, investigation of the literature shows these are unproven assumptions.

A- correct curvature localization is required for wild type cell shape.

(As mentioned in the response to reviewer 1.) “since 2004 that cells need multiple, long, and geometrically-localized MreB to form rods.”

While MreB might have a slight accumulation at negative curvature, no group, this one or any other, has shown curvature localization is causative of shape: MreB mutants have perturbed shapes, and perturbed curvature localizations. All of these points correlate, but they have not been uncoupled, perturbed curvature localization has not been shown to be causative. These mutants could be affecting many other properties of the filaments or its interactions, resulting in these shapes. (to claim this, one would have to make filaments localize to the wrong spot without affecting the filaments or any of the cell wall synthesis proteins).

We agree with the reviewer that no one has shown curvature localization causes shape. Indeed in this very paper we show that there are mutants with curvature localization that fail to have proper shape and conclude that curvature localization is not sufficient for shape. That said, the reviewer is correct that the fact that we have yet to identify a properly-shaped strain that lacks curvature localization does not formally prove that curvature localization is necessary for shape (we currently only have absence of evidence). Consequently, we have reworded our conclusions to focus on our specific findings: all properly-shaped strains have curvature localization but not all strains with curvature localization are properly shaped.

B- MreB curvature localization reflects increased sites of growth.

Lines 69-72 “In *E. coli*, the bacterial actin homolog MreB organizes cell wall insertion by localizing to regions of the cell with particular geometric curvatures and recruiting cell wall enzymes to direct growth to those sites”

This is completely false. While MreB does show a very slight (2%?) accumulation at given curvatures, no work has shown either 1) recruitment of cell wall enzymes to those site, Or More importantly 2), no work has shown that this accumulation leads to increased growth at those sites. Rather, recent work by the VanTeffelen group demonstrates curvature localization does not increase growth, a point not cited nor discussed.

We thank the reviewer for pointing out this poorly-worded sentence in the introduction, which we have revised. We respectfully point out that our paper’s conclusions (RodZ affects MreB polymer number/curvature localization and MreB polymer number/length/curvature localization are predictive of shape) simply do not depend on whether MreB reflects sites of cell growth. This is an interesting topic for future studies, and in the revised text we have both clarified the independence of our findings from this model and discussed future studies to clarify the mechanism by which MreB dictates shape. With respect to the final point about work from the VanTeeffelen group, those measurements were done on cells that had been mechanically strained and are thus growing quite differently from normal and that study did not examine cylindrical uniformity (the focus of our current work). Nevertheless, we have included a reference to their finding that geometric MreB localization may be insufficient to explain cell straightening in the introduction.

Lines 41-43 “Quantitative 3D measurements and a series of mutant strains show that among various properties of MreB, polymer number, total length of MreB polymers, and MreB curvature preference are the key determinants of cylindrical uniformity”

This data (part of which is suspect) shows corrections to cellular uniformity, not the determination of cellular uniformity.

We agree and have changed the word “determinants” to the more accurate “correlates”.

Lines 61-62 “ Previously we studied the determinants of *E. coli* straightening and population-average width, but the determinants of single-cell cylindrical uniformity remain unclear”

Here they reference their previous paper, which showed that the angle of filaments correlates with width. Again, because they could not independently modulate width, and measure angle, not angle, and width, the properties correlate, but they do not “determine” .

We agree and changed “determine” to “correlate” as well as rephrasing the title to better reflect this.

Lines 87-88 First, elongated polymers may coordinate the activity of cell-wall modulating enzymes to produce the twisting cylindrical growth seen in *E. coli*.

I’m shocked they use this reference as evidence for this statement. This work, by Wang, showed *E. coli* twist matched the twist of spirals of a problematic MreB fusion, whose spiral localization (and angle) was an artifact. This exact point was corrected by this group in a later work, that demonstrated a much more shallow angle. Why would they reference the angle of an artifactual filament structure as evidence for chiral growth, when the angle of cell twist no longer matches the real angle of MreB filaments?

Here we think the reviewer may be confusing the chirality of MreB filaments with the chirality of cell growth itself. Chiral growth of the cell occurs even in cells without MreB fusions; chirality is a general property of *E. coli* cell growth (see Fig. 3C from Wang which uses two WT strains and one A22 resistant strain). Our use of the reference is not to say that *E. coli* twist is the same as MreB polymer angles but that cells do indeed twist when growing. To help avoid confusion about what the reference is referring to, we have reworded this section and also point the reviewer to a more recent study on the role of MreB in chiral cell growth from the group of KC Huang.

REVIEWERS' COMMENTS:

Reviewer #3 (Remarks to the Author):

I agree with Reviewer 4 that in their original submissions the authors over-interpreted their data and glossed over some obvious differences in the curvature enrichment of MreB filaments in different mutant background. For example, the authors could be more explicit in judging whether a curvature enrichment curve is significantly different or similar to WT—all the curves, if judged by error bars, are significantly different from each other. Reviewer 4 is correct in that it is important not to confuse “correlation” with “determination”, as how the curvature dependent localization of MreB is related to cell shape determination is still an open question. With that said, however, I think the authors are also correct that their LASSO model can indeed predict what properties of MreB polymers are most correlated with cylindrical uniformity (does not imply a cause-effect relationship)— a random, non-relevant protein filament would not produce a high correlation. I think the current revision has tuned down significantly, and that most importantly, the observations presented in the study are valid. These results will provide interesting insight into the relationship between cell shape and MreB polymer properties, and the role of RodZ. As such, I believe that making these results available to the community will only promote further investigations and better understandings.